# A study of the morphology and effective density of externally mixed black carbon aerosols in ambient air using a size-resolved single-particle soot photometer (SP2)

Yunfei Wu[1*], Yunjie Xia[1, 2], Rujin Huang[3], Zhaoze Deng[4], Ping Tian[5], Xiangao Xia[4], Renjian Zhang[1*]

Key Laboratory of Regional Climate-Environment for Temperate East Asia (RCE-TEA), Institute of Atmospheric Physics, Chinese Academy of Sciences, Beijing 100029, China
University of Chinese Academy of Sciences, Beijing, 100049, China
Key Laboratory of Aerosol Chemistry and Physics, State Key Laboratory of Loess and Quaternary Geology, Institute of Earth and Environment, Chinese Academy of Sciences, Xi'an 710061, China
Key Laboratory of Middle Atmosphere and Global Environment Observation (LAGEO), Institute of Atmospheric Physics, Chinese Academy of Sciences, Beijing 100029, China
Beijing Weather Modification Office, Beijing 100089, China

*Correspondence to*: Yunfei Wu (wuyf@mail.iap.ac.cn) and Renjian Zhang (zrj@mail.iap.ac.cn)

**Abstract**

The morphology and effective density of externally mixed black carbon (*ext*BC) aerosols, important factors affecting the radiative forcing of black carbon, were studied using a tandem technique coupling a differential mobility analyzer (DMA) with a single-particle soot photometer (SP2). The study extended the mass-mobility relationship to large *ext*BC particles with a mobility diameter ($d_{mob}$) larger than 350 nm, a size range seldom included in previous tandem measurements of BC aggregates in the atmosphere. The experiment was conducted at an urban site in Beijing during a 19-day winter period from 23 January to 10 February 2018. Ambient dry particles were selected by the DMA, and the size-resolved *ext*BC particles were distinguished from particles with a thick coating (internally mixed) according to the time delay between the incandescence signal peak and the scattering peak detected by the SP2. The masses of the *ext*BC particles were then quantified. The time differences between the DMA

size selection and the SP2 measurement were processed previously. The normalized number size distributions were investigated at the prescribed $d_{mob}$ sizes in the range of 140–750 nm to provide the typical mass of $ext$BC at each $d_{mob}$. On this basis, the mass-mobility relationship of the ambient $ext$BC was established, inferring a mass-mobility scaling exponent ($D_{fm}$) (an important quantity for characterizing the morphology of fractal-like BC aggregates) with a value of 2.34±0.03 in the mobility range investigated in this study. This value is comparable with those of diesel exhaust particles, implying a predominant contribution of vehicle emissions to the ambient $ext$BC in urban Beijing. Compared to the clean period, a higher $D_{fm}$ value was observed in the polluted episode, indicating a more compact BC aggregate structure than that in the clean period. The effective densities ($\rho_{eff}$) of the $ext$BC in the same $d_{mob}$ range were also derived, with values gradually decreasing from 0.46 g cm$^{-3}$ at 140 nm mobility to 0.14 g cm$^{-3}$ at 750 nm mobility. The $\rho_{eff}$ values were slightly lower than those measured using the DMA-aerosol particle mass analyzer (APM) system. The difference in $\rho_{eff}$ values was likely due to the lower BC masses determined by the SP2 compared to those measured by the APM at the same mobility, since the SP2 measured the refractory BC (rBC) mass instead of the total mass of the BC aggregate, which consists of both rBC and a possible fraction of nonrefractory components measured by the APM. The $\rho_{eff}$ values in the 280–350 nm $d_{mob}$ range were much closer to the values for soot aggregates reported in the literature. It might be related to the more compact structure of BC aggregates in this range, resulting from the reconstruction effect by volatile and/or semivolatile components in the atmosphere. The reconstruction effect might also result in a hiatus in the increased dynamic shape factor in the range of 200–350 nm, which presented an overall increase from 2.16 to 2.93 in the 140–750 nm $d_{mob}$ range.

## 1 Introduction

Black carbon (BC), a byproduct of incomplete combustion, is the main light-absorbing component in atmospheric aerosols. BC can lead to positive radiative forcing second in magnitude only to $CO_2$ and thus warming of the earth's atmosphere (IPCC, 2013). However, there remains a large amount of uncertainty regarding the radiative forcing

induced by BC due to its complexity and variability in morphology, mixing state and hygroscopicity. Freshly emitted BC particles usually exhibit fractal-like aggregates composed of a number of primary carbon spherules (Park et al., 2004; Sorensen, 2011), which are generally hydrophobic. The condensation of organic and/or inorganic components leads to the collapse of fractal-like aggregates and, in turn, a compact structure of BC particles (Slowik et al., 2007; Zhang et al., 2008). Changes in the morphology of BC particles affect their optical properties. Encasement by organic and/or inorganic coatings also increases the absorption of BC particles through the lensing effect (Shiraiwa et al., 2010; Peng et al., 2016). In addition, water-soluble coatings increase the hydrophilic ability of BC particles (Zhang et al., 2008; McMeeking et al., 2011), indirectly affecting radiative forcing by affecting cloud processes.

Laboratory studies indicate that freshly emitted BC particles can become thickly coated within a few hours in the atmosphere (Pagels et al., 2009; Peng et al., 2016). Thus, many studies have focused on the optical properties and radiative forcing of thickly coated BC particles (Jacobson, 2001; Khalizov et al., 2009; Liu et al., 2017). However, *in situ* measurements have shown that a great number of uncoated and/or thinly coated BC particles exist in the ambient atmosphere, with a fraction even higher than that of aged BC particles (Schwarz et al., 2008). In general, thickly coated BC particles account for <50% of the BC-containing particles in urban areas based on single-particle soot photometer (SP2) measurements (Huang et al., 2012; Wang et al., 2014; Wu et al., 2016). The existence of a large fraction of uncoated and/or thinly coated BC particles is likely due to continuous emission from combustion processes such as vehicle exhaust (Wang et al., 2017). Therefore, studies on the radiative forcing of BC particles without thick coatings are also essential, especially in urban areas. First, the morphologies and sizes of these quasi-bare BC particles, which are the essential quantities for calculating the optical properties of BC particles in numerical models, should be investigated (Scarnato et al., 2013; Bi et al., 2013).

The morphology of fractal-like BC aggregates is generally characterized by a quantity called the fractal dimension ($D_f$), which has been well documented in the review

literature (Sorensen, 2011). The ideal diffusion-limited cluster aggregation (DLCA), to
which BC aggregates belong, has a $D_f$ value of $1.78\pm0.1$. Recent studies have also
reported a similar $D_f$ value of ~1.82 for bare soot particles using transmission electron
microscopy (TEM) analysis of aerosol samples collected in four different environments
(Wang et al., 2017). A significant increase in the $D_f$ was observed when the soot
particles were partly coated or embedded. In the past two decades, the morphologies of
BC aggregates have also been widely studied using tandem mobility techniques (Park
et al., 2008). Measurements obtained using an impactor (e.g., an electrical low-pressure
impactor (ELPI)) or a particle mass analyzer (e.g., an aerosol particle mass analyzer
(APM) or a centrifugal particle mass analyzer (CPMA)) connected in tandem with a
differential mobility analyzer (DMA) have revealed the relationship between particle
mass and mobility (Park et al., 2003; Maricq and Xu, 2004; Olfert et al., 2007; Rissler
et al., 2014; Sorensen, 2011; and associated references therein). The derived mass-
mobility scaling exponents ($D_{fm}$) which have also been called fractal dimensions in
some of these references, varied over a wide range of 2.2–2.8 for diesel exhaust
particles. These values were inherently higher than the virtual $D_f$, which is defined as
the scaling exponents between the radius of gyration of an aggregate and the radius of
primary spherules composing the aggregate, due to the improper interpretation of
mobility measurements, as demonstrated in detail in Sorensen (2011). The $D_f$ of diesel
particles obtained using TEM is ~1.75, corresponding to a large $D_{fm}$ value of ~2.35
based on the mass-mobility relationship (Park et al., 2004). The mobility size-
dependent effective densities ($\rho_{eff}$) of BC aggregates were also determined from the
DMA-ELPI or DMA-APM (or CPMA) measurements, which were difficult to
characterize using TEM techniques.
The previous tandem measurements generally provided the mass-mobility relationship
of particles with a mobility diameter ($d_{mob}$) not exceeding 350 nm due to the system
detection limit (Park et al., 2003; Maricq and Xu, 2004; Olfert et al., 2007; Rissler et
al., 2014). A condensation particle counter (CPC) is connected next to the DMA-APM
system to measure the number concentrations of mobility size-selected particles at
various APM voltages. The voltage is proportional to the particle mass, and the voltage
resulting in the maximum concentration is in turn considered the typical voltage of the
mass of particles with a prescribed mobility size. Because large particles (e.g., $d_{mob}$>350
nm) are less abundant in the atmosphere than smaller particles, larger uncertainties exist
in the DMA-APM-CPC measurements for the larger particles (Geller et al., 2006).
Hence, the extrapolation of the mass-mobility relationship established on the basis of
tandem measurements of small mobility diameters (e.g., $d_{mob}$<350 nm) to large particles
(e.g., $d_{mob}$>350 nm) is insufficient.
The SP2 was developed on the basis of the laser-induced incandescence technique and
provides advantages in the study of individual BC particle properties, including mass,
size and mixing state. The SP2 determines the refractory BC (rBC) mass from particle
to particle, thus providing the masses of BC aggregates throughout a wide size range
(70–500 nm mass-equivalent diameter according to the manufacturer) with high
sensitivity and accuracy (Schwarz et al., 2006). Recently, a tandem system consisting
of an SP2 connected to a DMA was developed to study the properties of size-resolved
BC aerosols in the atmosphere. The mass distributions and mixing states of the size-
selected BC were investigated in northern India using a DMA-SP2 tandem system
(Raatikainen et al., 2017). Coupling an SP2 with a volatility tandem DMA (VTDMA),
the rBC core size distributions of internally mixed BC and those measured by the
VTDMA were compared at the prescribed mobility size ranges. Subsequently, the
morphology and effective density of the internally mixed BC particles were studied
(Zhang et al., 2016). The hygroscopic properties of BC particles were studied using a
hygroscopicity tandem DMA (HTDMA)-SP2 coupling system (McMeeking et al., 2011;
Liu et al., 2013). Few studies have been performed on the morphology and effective
density of fractal-like BC aggregates that are not coated with other components,
especially those in the ambient atmosphere, usingDMA-SP2 measurements.
Using the DMA-SP2/CPC system, Gysel et al. (2012) revealed that the SP2 was unable
to reliably detect BC particles from a PALAS spark discharge soot generator due to the
lower detection limit of the SP2 for loosely packed agglomerates made up of small
primary spherules (~5–10 nm in diameter). However, they also claimed that a well-
aligned SP2 was expected to have a detection efficiency adequate to measure BC
aggregates (e.g., diesel exhaust soot) in the atmosphere because these BC aggregates
have larger primary spherules and substantially higher effective densities than the
agglomerates made up of small primary spherules. Therefore, in this study, a DMA-
SP2 tandem system was built to examine the mass-mobility relationship (from which
the morphology and effective density were further derived) of uncoated BC aggregates,
especially in the large particle size range (e.g., $d_{mob}$>350 nm), which has seldom been
included in previous tandem measurements. Moreover, the uncoated BC aggregates
were distinguished from the thickly coated BC particles using SP2, thus allowing the
study of the mass-mobility relationship of ambient BC aggregates in different
atmospheric environments. Previous DMA-ELPI or APM tandem measurements were
mainly conducted in the laboratory or in the source environments (e.g., in a tunnel)
where fresh BC aggregates were predominant.
Beijing, the capital of China, has suffered from severe air pollution issues in recent
years. Studies have revealed that emissions from coal combustion and/or biomass
burning for industry activities and residential heating have played a predominant role
in particulate pollution in Beijing, especially during the polluted episodes (Zhang et al.,
2013; Huang et al., 2014; Wu et al., 2017; Ma et al., 2017a, b). Thus, the variation in
the mass-mobility relationship of uncoated BC aggregates was also compared for a
polluted episode and a clean episode to examine the possible influence of a source
change on the morphology of these BC aggregates. In addition, a better mobility size
resolution (33 logarithmic size bins from 20 to 750 nm) was set for our DMA-SP2
system than was used in previous similar studies, in which only a few mobility
diameters in the range of ~150–350 nm were selected (Zhang et al., 2016; Liu et al.,
2013; McMeeking et al., 2011). Similar to the study presented by Raatikainen et al.
(2017), the high size resolution is advantageous for calculating the BC mass and
number size distribution in the polluted region in our future studies.

**2 Measurements**
**2.1 Experimental setup**
A tandem system comprising a size selection unit and a measurement section was built

and deployed in an ambient experiment that was conducted on the roof of a building (approximately 8 m above the ground) on the campus of the Institute of Atmospheric Physics, Chinese Academy of Sciences (IAP, CAS) during the winter from 23 January to 10 February 2018 (19 days in total). Located in an urban area of Beijing, the site is a few hundred meters from two main roads and thus may be significantly affected by vehicle emissions. More information on the measurement site is described in previous studies (e.g., Wu et al., 2016, 2017).

As shown in Fig. 1, polydisperse aerosols in the sample air were drawn through the size selection unit (a model 3087 neutralizer, a model 3080 classifier and a model 3081 DMA, TSI Inc., Shoreview, MN, USA) to generate quasi-monodisperse particles with a certain electrical $d_{mob}$. Before entering the system, the ambient air was dried by passing through a 12-inch-long Nafion dryer (model MD-700-12F-3, Perma Pure LLC, Toms River, NJ, USA). A vacuum pump was used to draw the dry sheath air (e.g., particle-free indoor air) opposite to the flow direction of the sample air to provide the appropriate vacuum degree required for the dryer. The size-selected particles were delivered to the measurement section for analysis by various methods, including an SP2 (Droplet Measurement Technologies, Boulder, CO, USA), a CPC (model 3776, TSI Inc., Shoreview, MN, USA) and two microaethalometers (model AE51, AethLabs, San Francisco, CA, USA). The operational flow rates were set to 0.1, 0.3 and 0.15 LPM (STP) for the SP2, CPC and two AE51s, respectively. The sheath flow rate was set to 3 LPM, resulting in a ratio of sheath-to-sample flow rate of 4.3:1 for the DMA. Particles in the range of 15–750 nm in mobility diameter could be selected. The flow rate for each instrument was calibrated using a soap film flowmeter (model Gilian Gilibrator-2, Sensidyne, Petersburg, FL, USA) before the experiment to ensure the accuracy of the selected particle sizes and measurements. The scientific purpose of this experimental setup was to study the mixing states of size-selected BC particles, the mass and number size distribution of BC, as well as the morphology and effective density of the uncoated BC aggregates that are discussed in the current study. Because only the DMA and SP2 were involved in the measurements presented in this study, the setting and operation of the two instruments were described and discussed in detail.

209

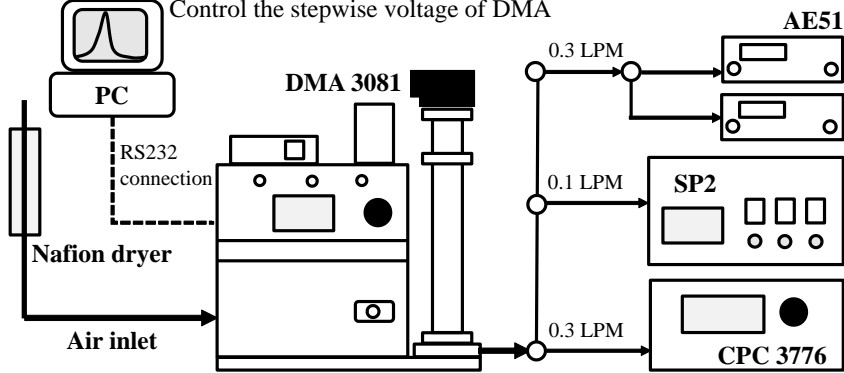

210

Fig. 1 Schematic of the experimental setup for size-resolved measurements of black carbon.

213

**2.2 Particle size selection**

The DMA was connected to an external computer on which a program was run to control the voltage of the DMA, i.e., the particle mobility diameter ($d_{mob}$). Thirty-three $d_{mob}$ values were set in the program to cyclically control the particles selected by the DMA and gradually increase from 20 nm to 750 nm on the logarithmic scale. Stepwise size selection was repeated until the operator stopped the program. A short cycle lasting for 18 s for each of the 33 diameters and a long cycle lasting for 36 s for each size were set to alternately operate in this experiment (Fig. S1 in the supplemental file). The purpose of these settings was to identify the time difference between the size selection and the subsequent measurement, as described in the following sections.

224

**2.3 Black carbon measurement**

The individual particulate rBC mass was measured by the SP2 according to the laser-induced incandescence signal when the particle passed through the intense Nd:YAG intracavity continuous laser beam (Schwardz et al., 2006) with a Gaussian distribution. The rBC mass in the SP2 detection range (~0.3–250 fg in this study, dependent on the laser intensity of a specific instrument) is proportional to the peak of the incandescence signal independent of the mixing state of the BC particles. If a BC particle is coated with nonrefractory components, the coating will evaporate before the rBC core

incandesces, leading to a time lag between the peaks of incandescence and scattering
signals that are synchronously detected by the SP2 (Moteki and Kondo, 2007).
According to the frequency distribution of the time lag, there was a significant
distinction between thickly coated (i.e., internally mixed) BC particles (*int*BC) and
thinly coated or uncoated (i.e., externally mixed) BC particles (*ext*BC) (Fig. S2) with a
minimum frequency at ~2 μs. BC-containing particles with delay times shorter than 2
μs were identified as *ext*BC. The delay time threshold might vary slightly from one SP2
to another; for example, Zhang et al. (2016) reported a short time lag of 1.6 μs. However,
the delay time threshold should be constant for a given instrument. In previous
measurements using the same SP2 employed in this study, the critical delay time was
maintained at 2 μs regardless of the ambient conditions, such as the pollution level (Wu
et al., 2016, 2017). A fraction of BC-containing particles with thin or even moderate
coatings might also be recognized as *ext*BC using the time delay approach (Laborde et
al., 2012). The effects of these thinly or even moderately coated BC particles are
discussed in Section 3.2 by reducing the delay time threshold from 2 μs to 1.2 μs and
0.4 μs, respectively.
The scattering signal of a single particle synchronously detected by the SP2 can be used
to estimate the optical size of the particle. The mixing state of a BC-containing particle
can be deduced by comparing the optical size of the particle and the mass-equivalent
size of the rBC core. Because the nonrefractory coating of a BC-containing particle is
evaporated due to the light absorption and heating of the rBC core when it passes
through the laser beam, the scattering cross-section of this particle, which is
proportional to the scattering intensity at a given incident light intensity, is gradually
decreased. To estimate the initial optical size of this particle, an approach called
leading-edge-only (LEO) fitting was developed (Gao et al., 2007). A small fraction of
the measured scattering signal in the initial stage before the particle is perturbed by the
laser is employed in the LEO fitting to reconstruct the expected scattering distribution
of the initial particle. In this method, the location of the leading edge in the beam is also
required, which is determined from a two-element avalanche photodiode (APD) signal.
Unfortunately, the notch in the two-element APD of our SP2 failed to fix in an adequate

position (e.g., before the peak location of the laser beam) in this experiment. Thus, the optical size and the consequent coating thickness of the BC-containing particle cannot be estimated. However, the coating thickness is not a crucial quantity in our current study on the morphology and density of uncoated BC aggregates. The coating thickness can provide a validation of our discrimination of *ext*BC but should have little influence on our final analysis and the discussion presented in the following sections.

Before the experiment, the incandescence signal was calibrated using DMA-selected monodisperse Aquadag particles. The effective densities of the mobility size-selected Aquadag particles were determined based on the polynomial equation as a function of the $d_{mob}$ reported in Gysel et al. (2011). The incandescence signal is more sensitive to the Aquadag particles than to ambient BC particles, because the Aquadag particle induces a higher incandescence signal peak (by a factor of ~25%) than fullerene soot or an ambient BC particle with the same mass (Laborde et al., 2012). Thus, the peak intensity of the incandescence signal was reduced by a factor of 25% when calculating the calibration coefficient. The calculated calibration factor, determined as the slope of the linear regression of rBC masses against the scaled peak heights of SP2's broadband incandescence signal, is consistent well with the factor estimated using a single-point scaling procedure (Baumgardner et al., 2012). The same calibration was performed again after the experiment. The calibration factors varied little (<3%), indicating the stability of the SP2 measurement during the entire experiment (Fig. S3). The uncertainty in the individual rBC mass determination is estimated to be ~10% due to the uncertainties in the rBC mass calibration and the effective density of the calibration material. An additional uncertainty may also arise in the determination of *ext*BC masses when the time delay approach is used to distinguish the mixing state of BC particles. The uncertainty will be further discussed in Section 3.2.

**3 Data processing**

**3.1 Identifying the time difference between the size selection and the SP2 measurement**

There exists a considerable difference between the time recorded by the size selection

program and that recorded by the SP2, due to the time cost of the particles transmitting from the DMA to the SP2, as well as the system clock difference between the computer on which the size selection program runs and that for the SP2 data acquisition. As shown in Fig. S1, the SP2 measurement occurs significantly later than the size selection. We have developed two methods to identify the time difference. The first method involves finding the time difference between the local peak in the particle number concentration (including both scattering and incandescence) detected by the SP2 and the beginning of the corresponding size selection cycle. During the experiment, stepwise size selection was cyclically performed to produce quasi-monodisperse particles with sizes gradually increasing from 20 nm to 750 nm. Thus, at the beginning of each new cycle, the voltage of the DMA should first drop drastically from a high value to a low one to make the particle size decrease from 750 nm to 20 nm. As a result, some particles with sizes in efficiently detectable range of the SP2 (~100–500 nm) are measured during the descent period, producing a local peak in the number concentration. Because it takes only a few seconds for the descent, identifying the occurrence time of the local peak position based on the SP2 clock and the beginning time of the size selection based on the external computer clock provides the time difference for each cycle.

The other method involves checking the consistency of the number and/or mass size distributions between the short-duration cycle and long-duration cycle. Although the durations of each size in the short cycle and long cycle are different (18 s vs. 36 s), the time difference between the size selection and the measurement should be uniform for adjacent short and long cycles. Setting an initial time difference and calculating the mean number and/or mass concentration of each particle size, the number and/or mass size distributions are obtained. Then, the correlation coefficients between the size distributions during short and long cycles are calculated. Changing the time difference gradually, we can obtain a set of correlation coefficients as functions of the time differences. The time difference resulting in the maximum correlation coefficient is considered the difference between the size selection and the measurement.

Since the detection efficiency of the SP2 decreases dramatically in the small particle

range (Fig. S4), the size distributions of the SP2-detected particles are inadequate for
further calculation of the correlation coefficients. Therefore, the former method was
employed in the current study to identify the time difference between the size selection
and the SP2 measurement. The latter method will be used to examine the time
difference between the size selection and the AE51/CPC measurements in our future
study on the number and mass size distributions of BC.

**3.2 Determination of the typical masses of extBC at prescribed mobility sizes**
Particles in a certain size range are selected by the DMA instead of absolutely
monodisperse particles in a given mobility size due to the effect of the transfer function.
In addition, larger particles with multiple charges are also selected. The frequency and
number size distributions of *ext*BC as a function of the mass-equivalent diameter of
rBC ($d_{me}$) at different mobility sizes are presented in Figs. S5 and S6, respectively. Note
that the number size distribution has been normalized by the peak value of the
corresponding distribution. Since the frequency and number size distributions of *ext*BC
are quite insufficient at small particle sizes ($d_{me}$<70 nm) due to the low detection
efficiency of the SP2 (Fig. S4), only the distributions with a $d_{mob}$ larger than 140 nm
are presented. In the following study, we mainly address the morphology and effective
density of *ext*BC in the 140–750 nm $d_{mob}$ range. The normalized number size
distributions at five representative $d_{mob}$ values (i.e., 140, 225, 350, 500, and 750 nm)
are also shown in Fig. 2. *Ext*BC particles with a considerable $d_{me}$ range were observed
for a certain $d_{mob}$, indicating a wide transfer function of the DMA due to the relatively
low ratio of sheath-to-sample flow (4.3:1). Multicharged particles also affected the size
distribution, especially in the $d_{mob}$ range of 100–400 nm (Ning et al., 2013). As shown
in Fig. S6 and Fig. 2, a minor peak is obviously observed in the right tail of the major
peak at each size distribution for $d_{mob}$ values of <350 nm.
As mentioned above, a fraction of thinly and/or moderately coated BC particles might
also be recognized as *ext*BC according to the time delay between the SP2 incandescence
and scattering signal peaks. These particles also have impacts on the size distribution
of *ext*BC for a given mobility size. A thinly-coated BC particle can be expected to have
a larger mass than a bare BC with the same mobility due to the restructuring of the
thinly coated BC particle by coating materials. These thinly coated BC particles will
increase the size distribution in the right tail when mixed with multicharged particles.
It is currently difficult or even impossible to separate the effects of the thinly coated
and multicharged particles based on the size distribution of *ext*BC. To examine the
possible effect of these thinly coated particles, we tightened the criterion of the delay
time for the discrimination of *ext*BC, gradually decreasing from <2.0 μs to <1.2 μs and
<0.4 μs. As shown in Figs. S5 and S6, a decrease in the delay time threshold results in
a significant reduction in the data volume used in the analysis but has few effects on
the major peak location of the distribution, which is used as the typical $d_{me}$ of *ext*BC
for a given mobility size. The typical $d_{me}$ values, determined as the mode values of the
lognormal function that are employed to fit the major peak of the size distribution at a
certain mobility size, vary little with the delay time thresholds (Table S1). The
maximum discrepancy in the $d_{me}$ is <3% throughout the prescribed mobility size range
in this study (140–750 nm). The delay time threshold-caused change mainly appears in
the right tail of the normalized number size distribution. Reducing the delay time
threshold to 0.4 μs results in a significant decrease in the fraction of particles with a
large $d_{me}$ compared to the 2.0 μs and 1.2 μs thresholds (Fig. S6). These large particles
are likely attributed to thinly and/or even moderately coated BC particles whose
structures are relatively more compact than the absolutely bare BC particles. Therefore,
we propose that thinly and/or even moderately coated BC and multicharged particles
should both have effects on the size distribution of *ext*BC, mainly in its right tail, but
have little influence on the typical $d_{me}$, which is considered as the peak $d_{me}$ of the
distribution for a given mobility size. The uncertainty in the typical $d_{me}$ due to the time
delay approach that was utilized to distinguish the *ext*BC is approximately 3% at a given
$d_{mob}$, which is in turn ~10% of the corresponding mass of *ext*BC. Combining the
uncertainty in the rBC mass determined by the SP2 (~10%), the total uncertainty in the
determined mass of *ext*BC should be ~20% in the studied mobility range of 140–700
nm. To achieve an adequate data volume for the analysis, the results and discussion
presented in the following sections are based on the database of *ext*BC discriminated as
BC-containing particles with delay times of less than 2.0 µs, unless otherwise specified.

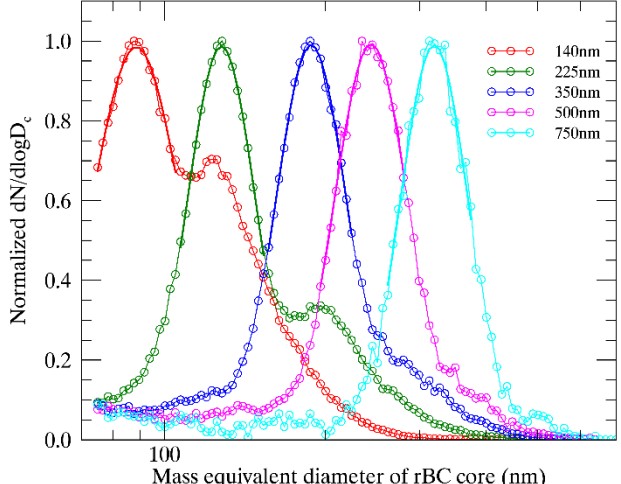


Fig. 2 Campaign average number size distribution of the mass-equivalent diameter of
the rBC core of *ext*BC normalized by the peak value at five representative mobility
diameters (140, 225, 350, 500 and 750 nm) selected by the DMA. Lognormal fitting is
performed for the major peak of each distribution.

**3.3 Theoretical calculation of the morphology and effective density**


The structure of *ext*BC, agglomerated by primary spherules with diameters of 20-60 nm
(Alexander et al., 2008), can be characterized by its mass-mobility relationship, which
is approximately expressed as a power law relationship between the mass of the
agglomerate particle ($m$) and its mobility diameter ($d_{mob}$), expressed as
$$m = k \cdot d_{mob}^{D_{fm}} \tag{1}$$
where the prefactor $k$ is a constant and $D_{fm}$ is the mass-mobility scaling exponent, which
was sometimes erroneously called the fractal dimension in previous studies (e.g., Park
et al., 2003). This quantity corresponds well to the virtual $D_f$ and represents the
morphology of the BC aggregates (Sorensen, 2011). The $D_{fm}$ value of a sphere is 3.
Thus, the morphology of a particle becomes increasingly closer to that of a sphere as
the $D_{fm}$ increases gradually to 3.
The effective density ($\rho_{eff}$) of the *ext*BC particles is calculated as the ratio of the BC
mass ($m$) measured using the SP2 and the BC volume, which is based on the $d_{mob}$
selected by the DMA, expressed as
$\rho_{eff} = \frac{6m}{\pi d_{mob}^3}$                           (2)
Combining Eqs. 1 and 2, $\rho_{eff}$ can also be expressed as a function of $d_{mob}$,
$\rho_{eff} = K \cdot d_{mob}^{D_{fm}-3}$                         (3)
where $K$ is a constant, corresponding to the prefactor $k$ in the mass-mobility relationship.
The dynamic shape factor is also calculated to indicate the morphology of the $ext$BC
particles. It is derived from the ratio of the slip-corrected mass-equivalent diameter ($d_{me}$)
and $d_{mob}$, expressed as
$\chi = \frac{d_{mob} \cdot C_c(d_{me})}{d_{me} \cdot C_c(d_{mod})}$                      (4)
where $d_{me}$ is calculated from the BC mass ($m$) by assuming the BC particle to be a
compact sphere with a density of 1.8 g cm$^{-3}$ (Taylor et al., 2015), and $C_c$ is the
Cunningham slip correction factor parameterized by particle diameter ($d$)
$C_c(d) = 1 + \frac{2\lambda}{d}[\alpha + \beta \exp(-\frac{\gamma \cdot d}{2\lambda})]$              (5)
where $\lambda$ is the mean free path of the gas molecules, which is set to 65 nm in this study
according to Zhang et al. (2016). The values of the three empirical parameters $\alpha$, $\beta$ and
$\gamma$ are 1.257, 0.4 and 1.1, respectively (Eq. 9.34 on page 407 in Seinfeld and Pandis,

421     2006).


**4 Results and discussion**
**4.1 Mass-mobility relationship of the ambient $ext$BC**
A power law relationship was applied to the $d_{mob}$-determined $ext$BC mass values,
delivering a campaign average mass-mobility scaling exponent ($D_{fm}$) of the ambient
$ext$BC (Fig. 3). In the $d_{mob}$ range of 140–750 nm, the fitted $D_{fm}$ is 2.34, with one standard
deviation of 0.03. The fitted $D_{fm}$ is close to the lower limit of the $D_{fm}$ values of diesel
exhaust particles presented in the literature, indicating the dominant contribution of
diesel exhaust to the $ext$BC in our measurement site in urban Beijing. Depending on the
fuel type, engine type and load, the $D_{fm}$ of diesel exhaust particles measured by the
DMA-APM or DMA-ELPI systems ranged between 2.22 and 2.84 (Olfert et al., 2007;
Maricq and Xu, 2004; Park et al., 2003 and references therein). The higher $D_{fm}$ values
in the literature are likely attributed to the higher fraction of volatile and/or semivolatile
components (e.g., sulfate) in the diesel exhaust (Park et al., 2003; Olfert et al., 2007).
The presence of these volatile and/or semivolatile components would result in a more
compact structure of the particle, leading to a higher $D_{fm}$ value for coated particles than
for bare BC aggregate. Because the rBC mass instead of the whole particle mass of
*ext*BC was measured by the SP2, a relatively low $D_{fm}$ value was expected and
reasonable in this study. In addition, the relatively low $D_{fm}$ value observed in urban
Beijing also likely implies high fuel quality (e.g., low sulfur content) and efficient
combustion in vehicle engines, which decrease the organic and/or inorganic fractions
in diesel exhaust particles. The $D_{fm}$ value for the ambient soot agglomerates measured
with a DMA-APM system near a diesel truck-dominated highway was 2.41 (Geller et
al., 2006), slightly higher than the value in our study.
According to Sorensen (2011), the ideal fractal-like DLCA with a virtual $D_f$ of
approximately 1.78 should have an expected $D_{fm} \approx 2.2$ in the slip flow regime in which
the BC aggregates are generally observed. The slightly larger $D_{fm}$ value of ambient
*ext*BC (2.34) in the current study might indicate a more compact structure than the ideal
fractal-like DLCA due to the reconstruction effect by other components in the
atmosphere. The reconstruction effect appears to be more significant in the smaller
particle range than in the larger particle range. The smaller BC particles are more likely
to be coated by volatile and/or semivolatile materials, which will be discussed in detail
in the next section. We piecewise fitted the mass-mobility relationship using the power
law function in the mobility ranges of 140–350 nm and 350–750 nm. A $D_{fm}$ of
2.51±0.04 that was obtained in the smaller mobility range (140–350 nm) was obviously
larger than the fitted value in the whole size range (140–750 nm). In contrast, a much
lower $D_{fm}$ with a value of 2.07±0.02 was observed in the larger mobility range (350–
750 nm). These results indicate that the ambient *ext*BC particles with larger mobility
diameters were likely less influenced by the reconstruction effect than those with
smaller mobility diameters.

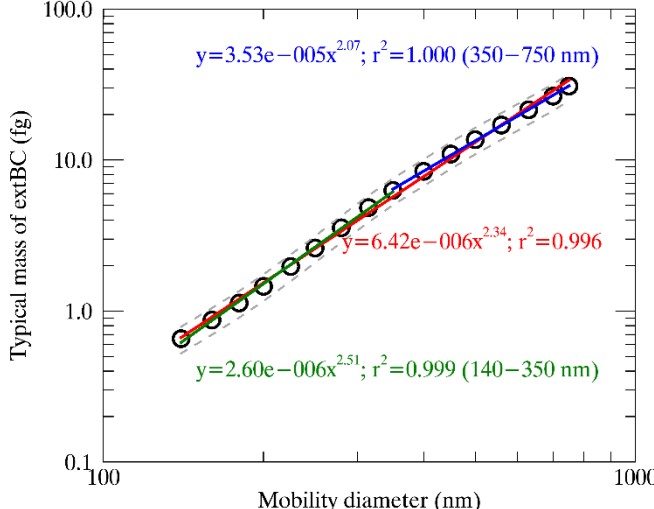


Fig. 3 The mass of *ext*BC particles as a function of the mobility diameter in the range
of 140–750 nm (black circles), fitted by a power law relationship (red line). The power
law functions piecewise fitted in the 140–350 nm mobility range (green line) range and
in the 350–750 nm mobility range (blue line) are overlaid. The dashed lines represent
the uncertainties in the determined *ext*BC masses.

Table 1 The typical mass-equivalent diameters ($d_{me}$) and corresponding masses of *ext*BC
for different mobility sizes ($d_{mob}$) selected by the DMA in the whole campaign, in a
polluted episode and in a clean period. The effective densities ($\rho_{eff}$) and dynamic shape
factors ($\chi$) at the $d_{mob}$ selected by the DMA throughout the whole campaign are also
presented.

| $d_{mob}$ (nm) | $d_{me}$ (nm) | | | mass (fg) | | | $\rho_{eff}$ (g cm$^{-3}$) | $\chi$ |
|---|---|---|---|---|---|---|---|---|
| | total | polluted | clean | total | polluted | clean | | |
| 140 | 88.8 | 87.2 | 88.5 | 0.66 | 0.63 | 0.65 | 0.46 | 2.16 |
| 160 | 97.5 | 96.9 | 98.1 | 0.87 | 0.86 | 0.89 | 0.41 | 2.27 |
| 180 | 106.2 | 106.1 | 107.0 | 1.13 | 1.13 | 1.15 | 0.37 | 2.35 |
| 200 | 115.6 | 116.1 | 115.5 | 1.46 | 1.48 | 1.45 | 0.35 | 2.40 |
| 225 | 127.9 | 128.6 | 128.4 | 1.97 | 2.01 | 1.99 | 0.33 | 2.41 |
| 250 | 140.5 | 142.2 | 141.0 | 2.62 | 2.71 | 2.64 | 0.32 | 2.41 |
| 280 | 155.8 | 158.0 | 154.4 | 3.56 | 3.72 | 3.47 | 0.31 | 2.41 |
| 315 | 172.6 | 174.8 | 170.6 | 4.85 | 5.04 | 4.68 | 0.30 | 2.40 |
| 350 | 188.2 | 191.8 | 185.9 | 6.28 | 6.65 | 6.05 | 0.28 | 2.41 |
| 400 | 207.4 | 213.7 | 207.4 | 8.41 | 9.20 | 8.41 | 0.25 | 2.43 |
| 450 | 226.4 | 232.3 | 225.9 | 10.94 | 11.81 | 10.87 | 0.23 | 2.50 |
| 500 | 243.8 | 251.4 | 242.2 | 13.65 | 14.98 | 13.39 | 0.21 | 2.62 |
| 560 | 262.6 | 271.1 | 260.1 | 17.06 | 18.77 | 16.58 | 0.19 | 2.71 |

| 630 | 283.2 | 293.5 | 282.5 | 21.42 | 23.83 | 21.25 | 0.16 | 2.81 |
| 700 | 305.1 | 312.7 | 305.0 | 26.76 | 28.83 | 26.73 | 0.15 | 2.89 |
| 750 | 319.6 | 328.8 | 323.5 | 30.76 | 33.49 | 31.92 | 0.14 | 2.93 |


The variation in the morphology of *ext*BC was further examined by comparing the mass-mobility relationship in a polluted episode with that in a subsequent clean period. As shown in Fig. S7, a polluted episode rapidly formed at 14:00 (local time, if not specified) on 26 January and lasted one and a half days to 0:00 on 28 January 2018. The mean $PM_{2.5}$ mass concentration was $72.1\pm23.1$ μg m$^{-3}$ in this polluted episode, three times the campaign average value ($23.0\pm26.7$ μg m$^{-3}$). The $D_{fm}$ value was $2.42\pm0.09$ in the polluted episode, higher than that ($2.33\pm0.06$) observed in the subsequent clean period from 1:00 on 28 January to 18:00 on 31 January 2018, during which the average $PM_{2.5}$ concentration was merely $8.9\pm2.7$ μg m$^{-3}$ (Fig. S8). The higher $D_{fm}$ in the polluted episode is mainly due to the increase in the masses of *ext*BC at large mobility sizes (e.g., $d_{mob}$ >250). As shown in Table 1, the typical masses of *ext*BC in the 280–700 nm $d_{mob}$ range in the polluted episode are ~7–13% larger than those in the clean period. Although the differences might result from the uncertainty (~20%) in the mass determination of *ext*BC, the commonly larger *ext*BC masses (in the 280–700 nm $d_{mob}$ range) to some degree still imply a possibly more compact structure of *ext*BC aggregates in the polluted episode, which might relate to changes in the dominant sources and the ambient environment. Previous studies have revealed that regionally transported pollutants emitted from coal combustion and/or biomass burning played an important or even predominant role in polluted episodes in Beijing (Wu et al., 2017; Ma et al., 2017a). Thus, a considerable fraction of *ext*BC aggregates from these sources is likely to coexist with the local vehicle-emitted BC aggregates in the polluted episode, even though the proportion of *ext*BC in the total BC-containing particles decreased (Fig. S9). These transported BC aggregates originating from coal combustion and/or biomass burning might have a more compact structure than those from vehicle exhaust due to the differences in the combustion environments and efficiencies. In addition, the BC aggregates might also become more compact due to the reconstruction effect by the

volatile and/or semivolatile components, which are generally abundant in polluted
episodes. Both possible factors are likely to result in the larger $D_{fm}$ values in the polluted
episode.

**4.2 Size-resolved effective densities of the ambient *ext*BC**
In contrast to the mass of *ext*BC ($m$), the effective density of the *ext*BC particles ($\rho_{eff}$)
showed a significant decreasing trend as the $d_{mob}$ increased from 140 nm to 750 nm
(Fig. 4 and Table 1). The highest $\rho_{eff}$ of 0.46 g cm$^{-3}$ was observed in the 140 nm $d_{mob}$,
likely because the BC aggregates at the smallest size are made up of the fewest primary
spherules. When the $d_{mob}$ increased to 750 nm, $\rho_{eff}$ decreased to as low as 0.14 g cm$^{-3}$,
approximately one-third of that at 140 nm. The very low $\rho_{eff}$ values agree well with the
fractal-like nature of the *ext*BC particles.

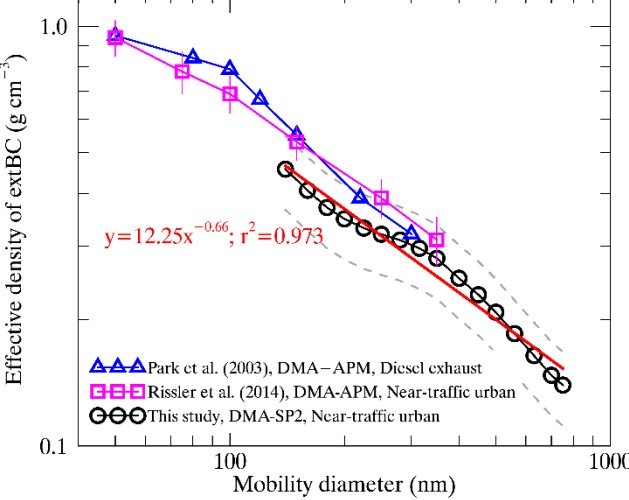


Fig. 4 The effective density ($\rho_{eff}$) of the *ext*BC particles as a function of the mobility
diameter ($d_{mob}$) (black circles). The red line represents the power-law fitting of $\rho_{eff}$
versus $d_{mob}$. The variations of $\rho_{eff}$ with $d_{mob}$ measured for the soot agglomerates from
diesel exhaust (Park et al., 2003) and near-traffic urban environments (Rissler et al.,
2014) are also presented as blue triangles and red squares, respectively. The dashed
lines represent the uncertainties in the determined $\rho_{eff}$.

The $\rho_{eff}$ values obtained by the DMA-SP2 measurements are close to those of the lower
limits of diesel exhaust particles measured by the DMA-APM (or CPMA) or DMA-
ELPI systems. Park et al. (2003) reported a decrease in the $\rho_{eff}$ of diesel exhaust particles
under a moderate (50%) engine load from 0.95 g cm$^{-3}$ to 0.32 g cm$^{-3}$ as the mobility
diameter increased from 50 nm to 300 nm (Fig. 4). The $\rho_{eff}$ values presented in Park et
al. (2003) are approximately 1.25, 1.18 and 1.05 times those in our study at ~150 nm,
220 and 300 nm in mobility diameter, respectively. The differences in $\rho_{eff}$ values
between our study and the literature are generally within the uncertainty (~20%) in the
mass determination of *ext*BC at prescribed mobility sizes. However, the commonly
lower $\rho_{eff}$ values are also likely due to the techniques used to determine the mass of BC
aggregates. Some previous studies on the $\rho_{eff}$ of diesel exhaust particles using the DMA-
APM or DMA-ELPI tandem measurements also showed a slightly larger $\rho_{eff}$ throughout
the comparable mobility ranges (e.g., ~150–350 nm) than that measured in this study
(Maricq and Xu, 2004; Olfert et al., 2007). The masses of the bare BC particles were
determined by the laser-induced incandescence technique of the SP2. In a previous
tandem system, the APM (or CPMA) or ELPI was utilized to determine the typical
mass of BC aggregates at a given mobility, and the BC aggregates are likely composed
of a fraction of volatile and/or semivolatile components in addition to the bare primary
particles. These volatile and/or semivolatile components increase the mass of the whole
particle, resulting in a larger $\rho_{eff}$ value for a certain mobility causing a compact structure
of the BC aggregate. For example, Olfert et al. (2007) found that the $\rho_{eff}$ of diesel
exhaust particles coated with minor sulfate and water contents (~2% of the total particle
mass) was ~0.4 g cm$^{-3}$ at 299 nm, only slightly larger than the value of diesel exhaust
particles (0.32 g cm$^{-3}$) measured in Park et al. (2003) and that of *ext*BC in the urban
atmosphere (0.31 g cm$^{-3}$) in our study at the same mobility size. However, the $\rho_{eff}$ value
increased significantly to ~0.71 g cm$^{-3}$ at a relatively high engine load of 40% due to
the high sulfate levels (~30% of the total particle mass) in the diesel exhaust particles
(Olfert et al., 2007).
The $\rho_{eff}$ values of ambient soot aggregates also showed a similar decreasing trend with
increasing $d_{mob}$ based on the DMA-APM system (Geller et al., 2006; Rissler et al.,
2014). Rissler et al. (2014) showed a decrease in the average $\rho_{eff}$ of BC aggregates from
0.94 g cm$^{-3}$ to 0.31 g cm$^{-3}$ in the near-traffic urban environment as the $d_{mob}$ increased
from 50 nm to 350 nm (Fig. 4), similar to that of the freshly emitted diesel exhaust
particles presented in Park et al. (2003). However, based on the same method, the $\rho_{eff}$
values of the ambient BC aggregates that mostly originated from diesel exhaust (Geller
et al., 2006) are substantially different from those presented in Rissler et al. (2014),
especially in the large particle size range. The $\rho_{eff}$ at ~350 nm was 0.17 g cm$^{-3}$ in Geller
et al. (2006), approximately half of the value presented in Rissler et al. (2014). The
reason for the discrepancy might be related to the large measurement uncertainties of
the DMA-APM system for large particles, e.g., with $d_{mob}$ sizes greater than 300 nm,
since these large particles are less abundant in the ambient atmosphere (Geller et al.,
2006). Compared to the results presented in Rissler et al. (2014), the $\rho_{eff}$ values of
ambient *ext*BC aggregates in our study are slightly lower, e.g., by ~17%, ~18% and ~6%
for $d_{mob}$ values of150 nm, 250 nm and 350 nm, respectively. The relatively higher $\rho_{eff}$
values are also likely attributed to the effects of volatile and/or semivolatile components
in the soot aggregates. Rissler et al. (2014) found that the residual mass fraction of
volatile and/or semivolatile materials in the soot aggregates was ~10%, even when the
sample air was heated to 300 °C before entering the system for measurement.
It is interesting to note that the $\rho_{eff}$ values appear to be closer to the values presented in
the literature using the DMA-APM measurements in the 280–350 nm $d_{mob}$ range (Fig.
4). As shown in Fig. 3, larger typical masses of *ext*BC in this range are also observed
beyond the logarithmic scaled linear curve that is fitted to the mass-mobility
relationship. The relatively larger masses and $\rho_{eff}$ values might imply a more compact
structure of *ext*BC aggregates in this range, which likely results from the reconstruction
effect by the ambient volatile and/or semivolatile components. As shown in Fig. S9, the
size-resolved number fractions of *ext*BC exhibit a minimum in the 280–350 nm $d_{mob}$
range, regardless of whether they are associated with the polluted episode or the clean
period. This finding indicates that particles in this mobility range are more likely to be
thickly coated by other components than are particles in the smaller or larger mobility
ranges. Zhang et al. (2016) also observed an increased coating thickness of the BC-
containing particles in the mobility range of 200–350 nm (Table 1 in the literature)
using the VTDMA-SP2 measurement at a suburban site ~70 km away from our
observation site, although the variation in the coating thickness in the larger mobility
range was not investigated. Notably, the number fraction of *ext*BC at each mobility size
presented in Fig. S9 is roughly calculated as the ratio of the *ext*BC number
concentration to the sum of *ext*BC and *int*BC, in which the multiply charged effects
were not corrected. Although the *ext*BC particles without coatings and/or with thin
coatings are the focus of the current study, the higher fraction of thickly coated BC
particles in the 280–350 nm $d_{mob}$ range implies a higher possibility that these *ext*BC
particles in the same range were affected by volatile and/or semivolatile materials in
the atmosphere, in turn resulting in a more compact structure of these BC aggregates.
Further detailed studies of the size distribution of BC (including *ext*BC, *int*BC and both)
and non-BC particles based on the combined measurements of SP2 and CPC are needed
in our further work to reveal the potential mechanism for this phenomenon.
Although the $\rho_{eff}$ of *ext*BC at small sizes ($d_{mob}$<140 nm) cannot be determined due to
the lower limit of the DMA-SP2 system, we extended the $\rho_{eff}$ of *ext*BC to a large size
range (350<$d_{mob}$<750 nm), which was barely investigated in previous studies using
tandem measurements. A continuous decrease in $\rho_{eff}$ with increasing $d_{mob}$ was observed
even in the large size range between 350 nm and 750 nm (Fig. 4). It is reasonable to
infer that the structure of the *ext*BC particles becomes looser when the fractal-like
aggregates built up by the primary spherules increase.

**4.3 Dynamic shape factors of the ambient *ext*BC**
Due to their fractal-like structures, the *ext*BC particles generally have large dynamic
shape factors ($\chi$) with values in the range of 2.16 to 2.93 (Table 1), much larger than
those of *int*BC with an average value of ~1.2 (Zhang et al., 2016). The $\chi$ value declined
exponentially as a function of coating thickness of BC-containing particles (Zhang et
al., 2016). In contrast to the decrease in $\rho_{eff}$, the $\chi$ values of *ext*BC generally increase as
$d_{mob}$ increases from 140 nm to 750 nm (Fig. 5). The *ext*BC particles 750 nm in mobility
diameter have a mean $\chi$ value of 2.93, approximately 1.36 times that for 140 nm $d_{mob}$
particles (Table 1). The larger particles have looser structures, resulting in higher $\chi$
values. However, the $\chi$ values appear to vary slightly in a narrow range between 2.40
and 2.41 in the size range of 200 nm to 350 nm (Fig. 5). The hiatus in the gradual
increase in $\chi$ is also likely related to the more compact structure of *ext*BC particles in
the 280–350 nm mobility range, which has been discussed in detail in the previous
sections.

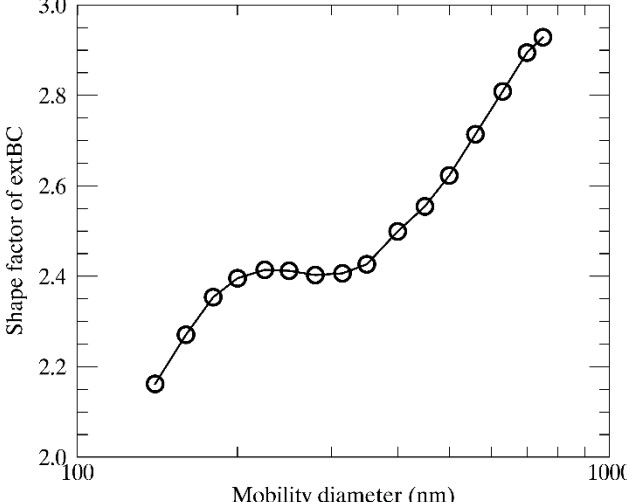


Fig. 5 The dynamic shape factor of the *ext*BC particles as a function of the mobility
diameter in the range of 140–750 nm.

**5 Conclusions**
The DMA-SP2 system was established to study the morphology and effective density
of the ambient *ext*BC particles, especially in the larger mobility size range, i.e.,
$350 < d_{mob} < 750$ nm, which was seldom investigated in previous tandem measurements.
Quasi-monodisperse particles in the $d_{mob}$ range of 20–750 nm were stepwise selected
using the DMA and then delivered to the SP2 for rBC mass measurement and mixing
state discrimination. The time difference between the size selection and the SP2
measurement was previously processed using the local peak approach. The normalized
number size distribution of *ext*BC, distinguished as having a delay time between the
incandescence signal peak and the scattering peak detected by the SP2 of less than 2 μs,
as a function of $d_{me}$ was investigated at each prescribed mobility size in the range of
140–750 nm. The size distributions at smaller mobility sizes were not presented due to
the lower limit of the rBC mass determined using the SP2. The peak $d_{me}$, calculated as
the mode value of a lognormal function fitted to the major peak of the size distribution,
was determined as the typical $d_{me}$ value at each mobility size. Consequently, the typical
mass of *ext*BC at each mobility size was identified. Reducing the time delay threshold
employed to discriminate the *ext*BC had few effects on the determined masses of *ext*BC,
implying the reliability of our study for *ext*BC particles. The uncertainty in the
determined *ext*BC masses was ~20%, based on a combination of the uncertainty in the
SP2-measured rBC mass and the uncertainty related to the time delay approach. On this
basis, the mass-mobility relationship of ambient *ext*BC in urban Beijing was
investigated. The campaign average $D_{fm}$ value was estimated to be 2.34±0.03 by fitting
the derived *ext*BC masses as a power law function of $d_{mob}$ in the range of 140–750 nm,
close to the lower-limit $D_{fm}$ value of diesel exhaust particles. A relatively larger $D_f$ value
was observed in the polluted episode than in the clean period (2.42±0.09 vs. 2.33±0.06),
implying a more compact structure of BC aggregates in the polluted episode.
A decrease in $\rho_{eff}$ with increasing $d_{mob}$ was observed, with the $\rho_{eff}$ value decreasing from
0.46 g cm$^{-3}$ at a $d_{mob}$ value of 140 nm to 0.14 g cm$^{-3}$ at 700 nm. The $\rho_{eff}$ values derived
using the DMA-SP2 measurement were slightly lower than those based on the DMA-
APM measurement. This difference was most likely due to the bias in the *ext*BC mass
determination using the SP2 and APM techniques. The pure rBC mass determined using
the SP2 in this study was generally lower than the total mass of the BC aggregate, which
comprises both rBC and a possible fraction of nonrefractory components. The $\rho_{eff}$
values in the 280–350 nm mobility range appeared to be much closer to the values for
soot aggregates reported in the literature by using the DMA-APM tandem measurement.
This finding might be related to the more compact structure of BC aggregates in this
range, which was likely influenced by the reconstruction effect of volatile and/or
semivolatile components in the atmosphere. The reconstruction effect might also result
in a hiatus in the gradually increased $\chi$ value in the range of 200–350 nm. Large $\chi$ values
generally increased from 2.16 to 2.93 with increasing $d_{mob}$, further implying the high
fractal structure of *ext*BC particles.

*Data availability*. Data used in this study are available from Yunfei Wu
(wuyf@mail.iap.ac.cn).

*Author contributions*. RZ led and designed the study; YW designed the study, set up
the experiment, analyzed the data, wrote and drafted the paper; YX and PT collected
the field data and contributed to data analysis; ZD provided the size selection procedure
and contributed to data analysis; RH and XX finalized the paper. All coauthors provided
comments on the paper.

*Competing interests*. The authors declare that they have no conflict of interest.

**Acknowledgments**
This work was supported by the National Key Research and Development Program of
China (nos. 2017YFC0209601 and 2017YFC0212701) and the National Natural
Science Foundation of China (nos. 41575150, 41775155, 91644217 and 91644219).

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
