# Peer review of "A study of the morphology and effective density of externally mixed black carbon aerosols in ambient air using a size-resolved singleparticle soot photometer (SP2)"

_Atmospheric Measurement Techniques, 2018_

## Referee Comment (RC1) · Anonymous Referee #1 · 7 Jan 2019

The paper by Wu et al. titled "A novel study of the morphology and effective density of externally mixed black carbon aerosols in ambient air using a size-resolved single-particle soot photometer (SP2)" presents measurements of the mass of rBC particles with known mobility diameters, sampled from the atmosphere of urban Beijing. A selected portion of the measurements (the most common masses for a given diameter) are interpreted in terms of two parameters commonly used in the soot community, the effective density and mass-mobility exponent (erroneously called the fractal dimension by the authors). The remainder of the measurements (the shape of the overall distribution) are not interpreted.

The manuscript as submitted represents an intelligent and detailed analysis of one aspect of the data set. However, it is not a complete analysis as discussed below, and the comprehensiveness of the work could be significantly improved. Moreover, the conclusions that the authors reach are in some ways predetermined by the analysis method. Therefore, the conclusions are misleading and the manuscript should be substantially revised. The data set shows significant promise, but before publication in AMT needs to be re-analyzed by asking the question, "what can we learn from these measurements?" instead of "how can we calculate common soot diagnostics from these measurements?"

As I explain below, the authors have accidentally only analyzed uncoated particles. It is not clear whether coated particles could be separated from multiply-charged particles using this method. Therefore, I have recommended rejection unless the authors can show that the problems of restructuring/coating and multiply charging can be separated. With this potential improvement, the paper might become a substantial contribution to the literature.

**MAJOR COMMENTS**

The authors have clearly thought carefully about their data and performed a careful analysis. The DMA was stepped instead of scanned, which avoids problems of data inversion otherwise associated with tandem DMA setups. My major comments are:

1. Limitations of the gaussian fitting

The authors have performed gaussian fitting to the number distributions of rBC-mass-per-particle measured by the SP2 after the DMA. But a huge part of this distribution is

not described by the gaussian fit. From my estimation about 50% of particles are not described, at smaller masses. This needs to be addressed quantitatively and seriously in the analysis.

The first hypothesis for the non-gaussian shape is multiple charging. The authors blame this on the distribution change in Section 3.2. This is possibly important. But also important would be restructuring due to coatings. Larger rBC particles can have smaller mobility diameters (Dm) after condensation of coatings (citations were given by the authors already). This would cause a tail to the right of the mode in Figure 2, as observed.

The hypothesis of coatings means that the authors' selection of the mode diameter resulted in their analysis of fresh, uncoated particles only. Therefore it is no surprise that the results indicate consistency with literature reports of fresh, uncoated particles. Therefore, the authors' results, conclusions and abstract must be rewritten.

It would be very interesting, for example, if the fraction of restructured particles could be separated from the fraction of multiply charged particles. This is also very difficult and may be impossible. I am not certain that it is impossible, but a very convincing argument would be required to show that the two problems could be separated.

I would like to note that the hypothesis of multiple charging would mean that smaller Dm should have a smaller fraction of total SP2 measurements explained by the gaussian fit (since there are more pre-existing particles available to become multiply-charged in the DMA). From my inspection of Figure 2 I do not see a strong trend with Dm. This makes me suspect that coatings are involved, but is not strong enough evidence for the authors to interpret the data as such.

On this topic two important related points should be made. Thick coatings are more likely to be acquired by smaller particles (Fierce et al., 2016). And it is possible that core-shell coatings are more likely for larger particles (Liu et al., 2017).

Fierce, L. et al. Black carbon absorption at the global scale is affected by particle-scale diversity in composition. Nat. Commun. 7:12361 doi: 10.1038/ncomms12361 (2016).

Liu, D. et al. Black-carbon absorption enhancement in the atmosphere determined by particle mixing state. Nat. Geosci., vol 10, pp 184–188 (2017). doi:10.1038/ngeo2901

2. Interpretation of the 'effective density' and 'fractal dimension'

The first major comment makes it clear that the 'effective density' and 'fractal dimension' results are biased towards fresh soot particles. In addition to this bias, the 'effective density' is a quantity which should correspond to the apparent density of a sphere with diameter equal to Dm. When using the DMA-SP2 setup employed in this study, the 'effective density' has virtually no meaning, since coatings are not measured by the SP2 as rBC. I do not see how this quantity could be useful for any future studies. If the authors wish to report such a quantity, they must explain in what context it should be interpreted. It should not be called 'effective density', which will confuse readers.

The quantity called 'fractal dimension' has the same problem as the 'effective density.' In addition, the quantity should have been called 'mass-mobility exponent' (Sorensen, Aerosol Sci Technol, 45:765-779, 2011, doi:10.1080/02786826.2011.560909).

It may be more interesting to compare the mixing state retrieved by asking 'is this particle similar to fresh soot?' (according to the 'effective density') with the mixing state retrieved by SP2 coating thickness analysis. But the usefulness of such an analysis is not guaranteed, the uncertainties may be too large.

Minor comments

Several minor comments:

1. It was not clear to me why the points in Figure S2 were quantized. Why does the number concentration nto vary smoothly?

2. In the abstract: effective density is not morpholgoy.

3. The discrepancy between DMA-APM and DMA-SP2 measurements cannot be explained by the SP2 only being sensitive to rBC. The SP2 is calibrated using an APM (or CPMA). The former DMA-APM studies used denuded soot. The discrepancy is due to the limitations of the SP2 calibration.

4. Line 55 – Thick and thinly coated needs to be defined. The authors may find that in fact most atmospheric BC is coated.

5. Line 88 – explain the reasons for the detection limits.

6. Line 110 – quasi-monodisperse needs to be quantified or specified precisely. AMT is a technical journal.

7. Line 149 specify the intensity of the laser in units of W/m2 or similar.

8. Line 153 frequency is the wrong word.

9. Line 185 define K.

10. Line 255 discuss the Diffusion-Limited Cluster-Cluster Aggregation mechanism (see Sorensen citation above)

11. Line 261 organics do not 'fill in the gap', they cause restructuring.

12. Line 357 the drag force, or uncertainty? Figure 6 needs error bars. Counting statistics will be poorer at the extreme sizes, possibly causing the observed trend.

---

## Referee Comment (RC2) · Anonymous Referee #2 · 7 Jan 2019

**\* General comments**

The authors use a tandem DMA-SP2 system to measure the mass-mobility exponent, effective density, and shape factor of ambient BC particles in Beijing, China. Despite the claim to novelty made in the title similar combined DMA-SP2 measurements have already been discussed and reported in the literature, which the authors have failed to discuss. In addition, there are significant shortcomings in the manuscript itself, ranging from inadequate description of the experimental details and results to data quality issues. In my view the manuscript is at the 'early draft' stage rather than the 'under

review' stage. I believe major revisions are required before the manuscript can be considered for publication in AMT. Most importantly, the authors need to better demonstrate why this study is an original contribution to the literature on the properties of ambient BC aerosols, and why their measurements should be considered artifact-free and trustworthy.

I see the following major issues:

The study is not placed in appropriate context through citation of relevant literature. In the title and elsewhere (e.g. L101) the authors claim that this is a 'novel' system for measuring the morphology and effective density of black carbon particles. This is incorrect. Tandem DMA-SP2 measurements have been discussed and/or performed in a number of different studies (e.g. Gysel et al., 2011; Raatikainen et al., 2017; Zhang et al., 2016). A handful of references are cited for more general mass-mobility measurements (e.g. using an APM rather than an SP2). But the literature on this topic is more extensive than this small selection of studies would suggest, including a review on the mobility of fracal aggregates by Sorensen (2011). I suggest the authors read this review and the references therein and put more effort into placing their measurements into the context of these previous works. In particular, the authors need to demonstrate what is the original contribution of this work.

The quality of the writing is not at a suitable level for scientific publication. There are many English grammar issues - too many to list in a scientific review. Beyond this, the language is frequently too vague. To take just the first example I come across the authors state that BC can lead to 'Earth warming' on L39. I believe that authors mean 'warming of the Earth's atmosphere' or similar. There are many more examples of such lazy language throughout the manuscript.

The study design and experimental and analytical details are inadequately described, which makes it difficult to judge their suitability. For example from what I can gather, the results presented in Figs. 2 to 6 are only for rBC-containing particles that displayed
a delay time of less than 2us (defined as 'extBC'). This needs to be clarified and stated more explicitly (another example of lazy language). The suitability of using delay time to distinguish between externally and internally mixed BC particles then requires further discussion. It is not as simplistic as the authors make it out to be. The authors allude to the fact that thinly-coated BC will exhibit low SP2 delay times on L280. This is also true for moderately-coated BC (e.g. with BC volume fractions as low as 30%), which would certainly not be classified as 'externally mixed' (ExtBC). A better approach for classifying BC mixing state with the SP2 is the quantitative LEO-fit approach (Gao et al., 2007; Laborde et al., 2012). The authors need to at least discuss this more sophisticated method, the reasons why they chose to use the more simplistic delay time approach, and the consequences of this decision.

The material as presented gives cause to question the quality of the measurement data, but insufficient details are provided to fully make this assessment. Specific points are highlighted below in relation to Fig. 2, which contains features that require explanation (absence of clearly defined peaks for multiply charged particles, presence of peaks with rBC mass approaching 0). In addition for reasons that are not yet satisfactorily explained, the effective density results presented in Fig. 5 are systematically lower than previous measurements (both the previous studies already cited by the authors as well as previous DMA-SP2 measurements that were also conducted in Beijing but are not yet discussed; Zhang et al., 2016). Given these issues, the authors should include further data and explanations to build confidence in their results and to confirm that the measurements are artifact-free. For example, were PSL spheres or other monodisperse particles (e.g. aquadag for the SP2 measurements) used to confirm that the DMA was operating correctly? What quality checks were performed to ensure the SP2 was operating correctly? (E.g. laser and detector block properly aligned, laser power levels, flow rate checks, comparison of calibration curves with previous calibration curves of the instrument).

No uncertainty estimates are provided for the measurements and main results. This is
important as the SP2 counting statistics and therefore measurement errors will be sizedependent. How sensitive are the reported quantities (e.g. Df) to these measurement uncertainties?

\* Specific comments

L69: 'Representativeness' instead of 'representation'

L80 - 82: The meaning of these sentences is not clear, rewording required.

L92: Schwarz misspelt.

L98: Statement needs qualification. This is not true when measuring pure BC particles.

L109: Please include information about the neutralizer that was used upstream of the DMA. This is especially required when considering the potential impacts of multiply charged particles as discussed in Section 3.2.

L135: Please include the length of the nafion dryer and the source of the dry sheath air.

L139: Some discussion is required about SP2 counting efficiency over this size range. It is not always 1, which will affect counting statistics, adding uncertainty to measurements reported at the limits of the range.

L153: The phrase 'frequency of the time lag' does not make sense. I guess the authors meant frequency distribution or histogram.

L156: Am I correct in assuming that the 'extBC' results presented later only include particles that displayed a lag time less than 2 us? If so please state this explicitly.

L162: More sensitive than what? To Fullerene soot I presume but statements like this need to be explicit, avoiding lazy language.

L166: Please include a Figure in the supplementary information of the two measured incandescence calibration curves. Were the incandescence calibrations performed all
the way up to 750 nm or were the calibration curves extrapolated? What sort of function was fit to the calibrations curves? Were the calibrations consistent with previous calibration curves measured for this instrument (an important check to make to ensure the SP2 was operating ok)

L176: Please also discuss what the prefactor k represents (e.g. Sorensen 2011).

L191: Please provide a reference for why this value of material density was chosen.

L199: The more common phrasing for this section would be 'Data processing'. And it seems to me that Section 2.4 would be a better fit in this section rather than the measurement methodology section above.

L220: It seems that this method was not used in this study. Why is it mentioned? Was it used to check the results processed with the first method?

L232: 'Minimizing the multicharged particles' is too vague. Suggest 'Correcting for the presence of multiply-charged particles' or something similar.

L240: Please be more specific. Exactly what parameter of the fitted curve was used to represent the mass of singly charged extBC particles?

Fig. 2: These curves contain features that require discussion to build confidence in the measurements. For example: related to the comment above about L156, do these frequency distributions only contain particles that display time lag less than 2us? If so, why are there sharp increases in the number of particles with rBC mass approaching 0 at diameters greater than 160nm? (With no filtering of the data applied I would assume these are heavily coated particles, which is a reason why I think the filtering process is not strict enough to be able to label these particles as 'extBC'). Why do the multiply-charged particles show up as a very fat tail that does not descend to a frequency of 0 until some point beyond the upper limit of the x-axis? What do these tails in the distribution represent? (E.g. they are in contrast to what is typically seen in SP2 calibrations, when doubly and triply charged particles are observed as clear, separate

AMTD
gaussian peaks). I suggest adding vertical lines along the x-axis corresponding to the mass of particles selected by DMA (e.g. under the assumption of spherical particles with the material density of BC) to provide reference points to compare the measured mass distributions to.

L252: Please provide further details for how the fit was performed. Were both the prefactor k and mass-mobility exponent Df allowed to vary freely? The k value seems to be very low in comparison to previous measurements (Sorensen 2011 and references therein), which requires explanation. How is the standard deviation of 0.04 in Df determined? How sensitive is the fitted Df to the size-dependent measurement uncertainties in rBC mass?

Fig. 3: Error bars are required in this figure to indicate the measurement uncertainties.

L263: This is a speculative statement that is not examined as thoroughly as it should be. If the sensitivity of Df to measurement uncertainty is taken into consideration is it still possible to conclude that the Df calculated here is 'relatively lower' than the previous measurements? If this result stands, can the authors provide any evidence to support their claim that fuel quality is higher in Beijing than for the studies they have cited?

L337: 'extBC' as defined with respect to delay time could also be comprised of BC mixed with small or even moderate amounts of non-BC material (e.g. L280). Additionally, the measurements reported here are lower than previous DMA-SP2 measurements of effective density in Beijing. Therefore, I find this explanation of the low effective density values measured in this study to be problematic.

\* References

Gao, R. S., Schwarz, J. P., Kelly, K. K., Fahey, D. W., Watts, L. A., Thompson, T. L., Spackman, J. R., Slowik, J. G., Cross, E. S., Han, J.-H., Davidovits, P., Onasch, T. B. and Worsnop, D. R.: A Novel Method for Estimating Light-Scattering Properties of
Soot Aerosols Using a Modified Single-Particle Soot Photometer, Aerosol Science and Technology, 41(2), 125–135, doi:10.1080/02786820601118398, 2007.

Gysel, M., Laborde, M., Olfert, J. S., Subramanian, R. and Gröhn, A. J.: Effective density of Aquadag and fullerene soot black carbon reference materials used for SP2 calibration, Atmos. Meas. Tech., 4(12), 2851–2858, doi:10.5194/amt-4-2851-2011, 2011.

Laborde, M., Mertes, P., Zieger, P., Dommen, J., Baltensperger, U. and Gysel, M.: Sensitivity of the Single Particle Soot Photometer to different black carbon types, Atmos. Meas. Tech., 5(5), 1031–1043, doi:10.5194/amt-5-1031-2012, 2012.

Raatikainen, T., Brus, D., Hooda, R. K., Hyvärinen, A.-P., Asmi, E., Sharma, V. P., Arola, A. and Lihavainen, H.: Size-selected black carbon mass distributions and mixing state in polluted and clean environments of northern India, Atmospheric Chemistry and Physics, 17(1), 371–383, doi:https://doi.org/10.5194/acp-17-371-2017, 2017.

Sorensen, C. M.: The Mobility of Fractal Aggregates: A Review, Aerosol Science and Technology, 45(7), 765–779, doi:10.1080/02786826.2011.560909, 2011.

Zhang, Y., Zhang, Q., Cheng, Y., Su, H., Kecorius, S., Wang, Z., Wu, Z., Hu, M., Zhu, T., Wiedensohler, A. and He, K.: Measuring the morphology and density of internally mixed black carbon with SP2 and VTDMA: new insight into the absorption enhancement of black carbon in the atmosphere, Atmospheric Measurement Techniques, 9(4), 1833–1843, doi:https://doi.org/10.5194/amt-9-1833-2016, 2016.

---

## Author Comment (AC1) · 5 Mar 2019

The paper by Wu et al. titled "A novel study of the morphology and effective density of externally mixed black carbon aerosols in ambient air using a size-resolved single particle soot photometer (SP2)" presents measurements of the mass of rBC particles with known mobility diameters, sampled from the atmosphere of urban Beijing. A selected portion of the measurements (the most common masses for a given diameter) are interpreted in terms of two parameters commonly used in the soot community, the effective density and mass-mobility exponent (erroneously called the fractal dimension

by the authors). The remainder of the measurements (the shape of the overall distribution) are not interpreted.

The manuscript as submitted represents an intelligent and detailed analysis of one aspect of the data set. However, it is not a complete analysis as discussed below, and the comprehensiveness of the work could be significantly improved. Moreover, the conclusions that the authors reach are in some ways predetermined by the analysis method. Therefore, the conclusions are misleading and the manuscript should be substantially revised. The data set shows significant promise, but before publication in AMT needs to be re-analyzed by asking the question, "what can we learn from these measurements?" instead of "how can we calculate common soot diagnostics from these measurements?"

As I explain below, the authors have accidentally only analyzed uncoated particles. It is not clear whether coated particles could be separated from multiply-charged particles using this method. Therefore, I have recommended rejection unless the authors can show that the problems of restructuring/coating and multiply charging can be separated. With this potential improvement, the paper might become a substantial contribution to the literature.

We greatly appreciate the reviewer for providing very constructive comments which have helped us improve the paper. We have considered the comments carefully and revised the manuscript accordingly, as detailed below in our point-to-point responses to the specific comments.

Major comments

The authors have clearly thought carefully about their data and performed a careful analysis. The DMA was stepped instead of scanned, which avoids problems of data inversion otherwise associated with tandem DMA setups. My major comments are:

1. Limitations of the gaussian fitting The authors have performed gaussian fitting to the

number distributions of rBC-mass-per-particle measured by the SP2 after the DMA. But a huge part of this distribution is not described by the gaussian fit. From my estimation about 50% of particles are not described, at smaller masses. This needs to be addressed quantitatively and seriously in the analysis.

The first hypothesis for the non-gaussian shape is multiple charging. The authors blame this on the distribution change in Section 3.2. This is possibly important. But also important would be restructuring due to coatings. Larger rBC particles can have smaller mobility diameters (Dm) after condensation of coatings (citations were given by the authors already). This would cause a tail to the right of the mode in Figure 2, as observed.

Reply: The particles selected by the DMA at a given voltage are generally quasi-monodisperse instead of having a single mobility diameter due to the transfer function of DMA. Thus, the peak diameter is used to represent the typical diameter of the particles selected by the DMA at the given voltage. In this study, we intend to obtain the typical mass of the externally mixed BC (extBC) particles selected by the DMA at a given voltage. Therefore, the masses corresponding to the peaks of the number distributions of rBC-mass-per-particle were required. In our original analysis, we investigated the frequency distribution of extBC mass at each prescribed mobility to obtain the peak extBC mass. However, the peak extBC mass directly obtained from the frequency distribution depends on the bin-size set in the frequency calculation. Thus, the Gaussian fitting was employed to identify the peak extBC mass with fewer artificial uncertainties.

In current study, only the peak locations of the frequency distribution of extBC mass are required. We don't concern about the practical shape of the distribution. In our original analysis, we also found that a considerable fraction of the extBC masses cannot be characterized by the Gaussian distribution, e.g., at the right tail of the distribution and the extremely small masses. The right tail beyond the Gaussian distribution can be partly interpreted by the effects of multi-charged BC particles as we discussed in the

manuscript. An extBC particle having double or more charges has a larger mass than the singly charged extBC. Fig. R1 shows the frequency distribution of the incandescence peak height values of Aquadag (a representative of bare rBC for the SP2 incandescence calibration) detected by the SP2, which are proportional to the rBC masses. The Aquadag particles were well mixed in the pure water and generated by an aerosol nebulizer, and then delivered to the SP2 for measurement after passing through a diffuse dryer. Thus, the Aquadag particles were mostly without coating and considered extBC. In this case, the doubly charged Aquadag particles exhibit a clear minor peak after the major one at a given mobility. The Guassian fitting for the major peak can distinctly separate the singly charged particles from those with double or more charges. The multicharged Aquadag particles account $\sim$50% of the total incandescence particles detected by the SP2 in the range of 140–250 nm in mobility diameter (dmob). The proportion gradually decreases to only a few percentage for particles larger than 600 nm dmob.

However, in the case of measurement for ambient particles, it is more complex. First, the time-delay approach used in this study can only distinguish the thickly coated BC particles from the BC particles with thinly or without coating. A fraction of thinly and/or even moderately coated BC particles cannot be absolutely separated from the bare BC particles. Thus, a fraction of extBC particles claimed in this study might be thinly-coated BC particles. We propose the right tail of the distribution is also associated with the thinly coated BC particles. These thinly-coated BC particles have relatively more compact structures, resulting in larger masses than the bare BC particles at a given mobility.

As for the distribution concentrated at the smaller extBC masses, we originally considered it was resulted from the detection noises of the SP2 for the smaller BC particles. As claimed by the manufacture, the lower BC detection limit is $\sim$0.3 fg per particle. Thus, data filter was performed by eliminating the particles with incandescence signal peak heights lower than 100. This criterion is determined from the calibration curve

by using Aquadag particles (Fig. S3 in the revised supplemental file). However, the distributions concentrated at the smaller masses still exist even when the data filter is processed. Initially, we suspected the smaller masses belonged to the thickly coated BC particles which were not separated from the extBC particles using the time-delay method. However, we found that the smaller masses also existed for the bare BC particles (i.e., Aquadag). As shown in Fig. R1, the particles with these smaller masses accounted for only 1∼9% of the total particle number (after data filter by eliminating the signal with incandescence peak value lower than 100) in the 140–200 nm dmob range. This proportion gradually increased to ∼37% at 700 nm mobility. Thus, these smaller masses should not be resulted only from the 'remnant' BC particles with thick coatings. We still attribute these smaller rBC masses to the measurement noises of the SP2.

In our revised manuscript, we make an improvement in data processing by investigating the number size distribution of the rBC core in extBC particles (dN/dlogDc, where Dc is the mass-equivalent diameter of rBC core by assuming it is a void-free sphere with 1.8 g cm-3 density, N is the number concentration in a given Dc range) instead of the frequency distribution of extBC mass at each prescribed mobility (Fig. 2 and Fig. S6 in the revision). This improvement would reduce the artificial uncertainties in the determination of extBC masses roughly obtained from the frequency distribution. The effect of detection noises at the small rBC masses also appears to be weakened by using the number size distribution. The size distributions at different mobility sizes are generally used in the studies of size-resolved BC particles. For instant, Zhang et al. (2016) measured the morphology and density of internally mixed BC (In-BC for short in the reference) with SP2 and VTDMA. They used the peak diameter of the normalized volume size distribution of the In-BC core to represent the typical diameter of In-BC core at the prescribed dmob selected by the DMA.

As shown in Fig. 2 and Fig. S6 in the revision, a minor peak at the right side of the major peak can also be observed at a given mobility, especially in the range with dmob smaller than 400 nm. The minor peak is related to both the multiply charged extBC

and the thinly coated BC particles which were also recognized as extBC by using the time-delay discrimination. To examine the effect of delay time threshold employed to discriminate extBC on the determined typical extBC mass, normalized number size distributions of extBC distinguished with delay times < 2.0 $\mu$s (red), <1.2 $\mu$s (green) and <0.4 $\mu$s (blue) are compared (Fig. S6 in the revised supplemental file). Reducing the delay time threshold results in a significant reduction in the data volume that used to calculate the distribution (Fig. S5 in the revised supplemental file) and also a reduction in the extBC number fraction at the right tail of the size distribution (more significant when delay time threshold decreases to 0.4$\mu$s). Reducing the delay time threshold seems to affect little on the peak location of the size distribution, which is considered the typical mass-equivalent diameters (dme), in turn the typical mass of extBC at a given mobility. The discrepancies in the lognormal-fitted peak dme by using different delay time thresholds don't excess 3% in the current study. Similar discussion is presented in detail in Section 3.2 of our revised manuscript. It implies the typical dme of extBC obtained from the peak dme of the number size distribution of the rBC core in extBC particles at each mobility is reliable in our study.

Fig. R1 Frequency distributions of incandescence peak height detected by the SP2 for size-selected Aquadag particles at the prescribed mobility sizes. The Gaussian fitting is performed for the major peak of each distribution to obtain the typical incandescence peak height at the corresponding mobility size.

The hypothesis of coatings means that the authors' selection of the mode diameter resulted in their analysis of fresh, uncoated particles only. Therefore it is no surprise that the results indicate consistency with literature reports of fresh, uncoated particles. Therefore, the authors' results, conclusions and abstract must be rewritten.

Reply: The initial purpose of our experimental setup is to study the microphysical and optical properties of size-resolved BC aerosols, including the sizes, mixing states and their impacts on light absorption at different mobility sizes, as well as the morphology and effective density of extBC aggregates that presented in the current study. Similar

experimental setup for the measurement of ambient BC particles can also be found in the recent literature (e.g., Zhang et al., 2016; Raatikainen et al., 2017).

Indeed, the morphology of BC aggregates has been widely studied by using different techniques, including transmission electron microscopy (TEM) and tandem measurements (e.g., DMA-APM). The morphology of diffusion-limited cluster aggregation (DLCA), to which the BC aggregates belong, has even been well documented in the review literature (Sorensen, 2011). However, as mentioned in our revised introduction, previous tandem measurements generally investigated the morphology and density of BC aggregates at small mobility sizes with the dmob not exceeding 350 nm. It is likely due to the large uncertainties in the measurement for larger particles which were less abundant in the atmosphere. In our study, we use a DMA-SP2 tandem system to study the mass-mobility relationship in a much larger mobility size range (140–750 nm) based on the sensitive and accurate SP2 measurement. The morphology and effective density determined in the relatively larger size range (140–750 nm) is compared to those in the smaller range (∼50–350 nm) in the literature to examine the applicability of mass-mobility relationship established at the smaller range in previous tandem measurements. In addition, variation in the morphology and effective density of ambient extBC aggregates in the atmosphere is also studied by comparing the mass-mobility relationship in a polluted episode to that a clean period. Although the DMA-SP2 is not a novel system (Gysel et al., 2011, 2012; Zhang et al., 2016; Raatikainen et al., 2017; and other associated references cited in our revised manuscript), this tandem system is seldom used to study the microphysical properties of ambient extBC aggregates which are also abundant in the atmosphere. Especially, similar studies are deficient in urban Beijing, where the particulate pollution is severe in recent years.

We have modified our manuscript substantially. Almost all the sections are rewritten.

It would be very interesting, for example, if the fraction of restructured particles could be separated from the fraction of multiply charged particles. This is also very difficult and may be impossible. I am not certain that it is impossible, but a very convincing

argument would be required to show that the two problems could be separated.

Reply: It is very difficult or even impossible to separate the fraction of restructured particles from the fraction of multiply charged particles. We have no idea how to deal with this issue, at least at the current stage. However, this problem should not have substantive impact on the results presented in current study. As mentioned above, the peak dme (i.e., mass) of the number size distribution of the rBC core in extBC particles were required and considered the typical dme (i.e., mass) of extBC at the prescribed mobility. Both the fractions of restructured particles due to thin coating and multicharged particles should only affect the shape of the distribution, e.g., cause an increase in the number size distribution at its right tail. They have few impacts on the peak location of the distribution as presented above by examining the effect of delay time threshold chosen to discriminate extBC particles.

I would like to note that the hypothesis of multiple charging would mean that smaller Dm should have a smaller fraction of total SP2 measurements explained by the Gaussian fit (since there are more pre-existing particles available to become multiply-charged in the DMA). From my inspection of Figure 2 I do not see a strong trend with Dm. This makes me suspect that coatings are involved, but is not strong enough evidence for the authors to interpret the data as such.

Reply: The frequency distribution of the rBC mass in extBC particles has been replaced by the size distribution, because the frequency distribution is too rough to presented in the scientific manuscript while the size distribution (dN/dlogD) is more practical. As presented in the responses above, we have explained the reasons of the high frequency at smaller rBC masses and attributed it to the measurement noises of the SP2. These smaller rBC masses influence little of the normalized number size distribution. Meanwhile, the increase in the size distribution at its right tail is interpreted as the co-effects of multicharged extBC and thinly coated extBC particles which cannot be absolutely separated from the bare BC aggregates using the time-delay discrimination. During the data reanalysis, we also calculated the normalized size distributions of rBC core

[Figure]

in thickly coated BC particles at the prescribed mobility sizes. As shown in Fig. R2, the rBC core of thickly coated BC (intBC for short) particles exhibit generally lower mass-equivalent diameters than extBC particles at each dmob. Note that the size distributions with dmob<200 nm are not presented due to the lower detection limit of the SP2. It can be expected because the mobility of the whole intBC particle, which is composed of a rBC core and a considerable non-refractory materials, was selected by the DMA while only the mass of rBC core was measured by the SP2. For extBC, the mass of particle is mostly attribute to the rBC core.

Fig. R2 Normalized number size distributions of the rBC core in extBC (red) and thickly coated BC particles (magenta)

On this topic two important related points should be made. Thick coatings are more likely to be acquired by smaller particles (Fierce et al., 2016). And it is possible that core-shell coatings are more likely for larger particles (Liu et al., 2017).

Reply: We have read the two suggested references carefully. However, only the extBC aggregates without coating and a possible fraction of thinly coated BC aggregates are concerned in current study. The size-resolved mixing state of BC-containing particles will be discussed in our future study. In the revision, rough mixing states of size-resolved BC particles, calculated as the ratios of number concentration of extBC particles to the sum of extBC and intBC particles, at different mobility sizes are presented as shown in Fig. S9. Note that effects of multicharged particles are not eliminated. However, the multicharged particles should result in a similar effect on extBC and intBC at a given mobility. Thus, they should influence little of the number fraction of extBC in the total BC-containing particles. Size distribution of number fraction of extBC is used to auxiliary interpret the possible mechanism of the relatively higher masses and effective densities of extBC in the dmob range of 280–350 nm in the revision (Lines 566–591).

2. Interpretation of the 'effective density' and 'fractal dimension' The first major comment makes it clear that the 'effective density' and 'fractal dimension' results are biased towards fresh soot particles. In addition to this bias, the 'effective density' is a quantity which should correspond to the apparent density of a sphere with diameter equal to Dm. When using the DMA-SP2 setup employed in this study, the 'effective density' has virtually no meaning, since coatings are not measured by the SP2 as rBC. I do not see how this quantity could be useful for any future studies. If the authors wish to report such a quantity, they must explain in what context it should be interpreted. It should not be called 'effective density', which will confuse readers. The quantity called 'fractal dimension' has the same problem as the 'effective density.' In addition, the quantity should have been called 'mass-mobility exponent' (Sorensen, Aerosol Sci Technol, 45:765-779, 2011, doi:10.1080/02786826.2011.560909). It may be more interesting to compare the mixing state retrieved by asking 'is this particle similar to fresh soot?' (according to the 'effective density') with the mixing state retrieved by SP2 coating thickness analysis. But the usefulness of such an analysis is not guaranteed, the uncertainties may be too large.

Reply: Thanks for the professional comments. As presented in the original manuscript, the effective density and fractal dimension of externally mixed BC (extBC) were analyzed instead of the entire BC-containing particles. The morphology and effective density of thickly coated BC particles were not discussed in our study. They have been studied in the previous literature, e.g., Zhang et al. (2016), using the SP2 and VTDMA measurements. In our study, the extBC particles are identified according to the time-delay between the incandescence signal peak and the scattering signal peak measured by the SP2. Only the BC-containing particles with delay times less than the criterions (2.0 $\mu$s or even lower) are considered extBC particles. The remaining BC-containing particles are considered thickly coated by nonrefractory components. As mentioned in our responses to the first major comment, a fraction of thinly coated BC particles might also be considered the extBC particles based on the time-delay approach. These thinly coated BC particles should influence the shape of the frequency or size distribution but have few impacts on the determined typical mass or mass-equivalent diameter of the

extBC. Thus, we propose that the results presented in our current manuscript should be reliable.

Actually, the thickness of the BC-containing particle can be retrieved using a specific approach, i.e., lead-edge-only (LEO) fitting (Gao et al., 2007). The method can estimate the optical size of particles using the scattering signal detected by SP2 at the beginning stage (e.g., the first 5%) by assuming certain parameters, including the refraction indices of coating materials and the rBC core. Compared to the size of the rBC core determined using the incandescence signal, the thickness of the coating can be estimated. However, this approach needs the scattering signal measured by the split channel of the SP2 (a two-element APD). The scattering signal detected by this channel is used to determine the center location of the Gaussian-distributed laser beam. Unfortunately, this channel of our SP2 was out of work during the experiment. We failed to use this method to estimate the coating thickness of individual BC-containing particle in the present study. However, the 'coating thickness' is not a crucial quantity in our current study. Only the extBC particles without or possibly with thin coating are investigated. Moreover, because a number of assumptions are employed in the LEO fitting, large uncertainties also exist in the retrieved coating thickness. Similar statements have presented in our revised manuscript as shown from Line 248 to 267.

Since only the extBC particles are analyzed, the 'effective density' and 'fractal dimension' should have practical meanings. Actually, the 'fractal dimension' we defined on the basis of the mass-mobility relationship of extBC is the 'mass-mobility scaling exponent'. It is different from the virtual 'fractal dimension' of BC aggregates (Sorensen, 2011), while also represents the morphology of BC aggregates. Although the 'effective density' and 'mass-mobility scaling exponent' of fresh soot particles were measured using a range of methods, the DMA-SP2 method was seldom used, especially in the ambient atmosphere. The advantage of the DMA-SP2 method is that the SP2 can distinguish the extBC from the thickly coated BC particles at the same time as measuring the masses of their rBC core, allowing to obtain the mass-mobility relationship in

different atmospheric environments. While previous studies using the DMA-APM-CPC method were mainly conducted in the laboratory or in the source environments (e.g., in the tunnel). Meanwhile, the DMA-APM-CPC generally measured the mobility size not exceeding 350 nm in dmob due to the very number concentrations of soot particles at larger particle sizes with the dmob>350 nm. The SP2 measures the particle-to-particle rBC mass, thus has a higher efficiency in the detection of larger BC-containing particles. The stepwise measurement of DMA-SP2 also provide a high time-resolution of the 'effective density' and 'mass-mobility scaling exponent' at different mobility size, which can be used to study the variation of these quantities and the possible reasons. Similar statements are presented in the last paragraph of the introduction in the revision.

In addition, we find that the effective densities of size-resolved extBC obtained from the DMA-SP2 measurement are slightly lower than those determined using the DMA-APM-CPC method. Although we have given reasonable interpretation to the difference in our manuscript, further measurements are needed to examine the difference between the two tandem methods, e.g., by measuring diesel exhaust particles synchronously. The uncertainty of the mass determination of extBC is estimated to be ∼20% at each dmob, including the uncertainty (∼10%) in the rBC mass which is converted from the incandescence peak height measured by the SP2 and the uncertainty (∼10%) arisen from the discrimination of extBC by using different delay time criterions. We have added similar discussion on the measurement uncertainties in the revision (Lines 277–283, 371–376).

Minor comments: 1. It was not clear to me why the points in Figure S2 were quantized. Why does the number concentration nto vary smoothly?

Reply: Figure S2 (Fig. S1 in the revised supplemental file) showed an example of the number concentrations of particles (including scattering and incandescence ones) detected by SP2 during a short circle and a long one. The operated flow rate of the SP2 was set to 100 cc per minute during this experiment. The smooth number concentration

is related to the quite low number concentration of the size-selected particles.

2. In the abstract: effective density is not morphology.

Reply: We have modified. We have checked throughout our manuscript to make the expression more rigorous. Moreover, we have written the abstract of the revised manuscript.

3. The discrepancy between DMA-APM and DMA-SP2 measurements cannot be explained by the SP2 only being sensitive to rBC. The SP2 is calibrated using an APM (or CPMA). The former DMA-APM studies used denuded soot. The discrepancy is due to the limitations of the SP2 calibration.

Reply: We calibrated our SP2 using the size-resolved Aquadag particles selected by the DMA as mentioned in the section 2.3 of our revision manuscript. The effective densities of mobility size -selected Aquadag particles were referred to Gysel. (2011). Calibrations were well conducted before and after the experiment. As shown in Fig. S3 in the revised supplemental file, the calibration curve obtained after the campaign is consistent to that obtained before the campaign (with difference in the calibration factor < 3%). Even when the denuded soot particles are measured, there also remain a possibly small fraction of non-BC components which can be measured by the APM but cannot be determined by the SP2. Rissler et al. (2014) revealed that the residual mass fraction of volatile and/or semivolatile materials in the soot aggregate was ∼10% even when the sample air was heated to 300 °C before entering the system for measurement.

4. Line 55 – Thick and thinly coated needs to be defined. The authors may find that in fact most atmospheric BC is coated.

Reply: We used the time-delay method to distinguish the mixing state of refractory BC (rBC). Two type rBC-containing particles can be distinguished. One is the rBC thickly coated by nonrefractory components and the other is rBC thinly coated or without coating (bared). The thickness of the coatings cannot be accurately identified in current study. As mentioned in the introduction of the manuscript, there also be a considerable fraction of extBC in the atmosphere especially in the urban regions (Lines 77–82 in the revision). The extBC discriminated using the time-delay method accounted for even >50% of the total BC-containing particles which were detectable by the SP2 in urban Beijing, although a fraction of extBC might be thinly or even moderately coated. The TEM analysis also showed that the bare-like soot particles accounted for 25% of the total soot particles in urban, and as high as 64% in tunnel. 63% soot particles were partly coated in urban environment (Wang et al., 2017). Thus, the morphology and effective density of these extBC aggregates in the atmosphere are studied.

Actually, the thickness of the coat can be retrieved from the scattering signals of the SP2 using the lead-edge-only (LEO) fitting (Gao et al., 2007). However, the notch in the two-element APD of our SP2 failed to fix in an adequate position in this experiment. Thus, the optical size and the consequent coating thickness of the BC-containing particles cannot be estimated in current study. However, the coating thickness is not a crucial quantity in our current study on the morphology and density of uncoated BC aggregates. It can provide a validation of our discrimination of extBC but should have little influence on our final analysis and discussion presented in the current study. Similar statements are presented in Lines 248–267 in our revised manuscript.

5. Line 88 – explain the reasons for the detection limits.

Reply: As mentioned in Lines 111–123 in the revision, larger uncertainties exist in the DMA-APM-CPC measurement for larger particles (e.g., with a dmob >350 nm) which are much less abundant in the atmosphere than the smaller particles. Thus, the previous tandem measurements generally provided the mass-mobility relationship of BC aggregates with a dmob not exceeding 350 nm.

6. Line 110 – quasi-monodisperse needs to be quantified or specified precisely. AMT is a technical journal.

Reply: Due to the effect of DMA transfer function, particles within a certain dmob range are selected by the DMA at a given voltage. Usually, the particle number presents a triangle distribution as the function of dmob at a given voltage. The peak of the triangle distribution corresponds to the certain dmob which we are used in this study. A slight modification to the statement is performed in the revision (Line 184). In the current manuscript, we won't specify the principle of DMA that used to quantify the quasi-monodisperse since it is out of our current research scope.

7. Line 149 specify the intensity of the laser in units of W/m2 or similar

Reply: Usually, the intensity of the laser is not specified in the SP2 study. Although the intensity of laser would influence the detection efficiency of SP2, it is usually checked indirectly by using the monodisperse Polystyrene Latex (PSL) at a given size (e.g., 269 nm). The intensity of the laser depends on the laser current which can be set manually in the SP2 acquisition software.

After the experiment, we used the DMA-SP/CPC system to measure the bare BC (Aquadag) particles. The detection efficiency was obtained by comparing the particle number concentration detected by the SP2 to that measured by the CPC at each prescribed mobility, as shown in Fig. S4 in the revised supplement file. Adequate detection efficiencies ($> \sim 90\%$) of SP2 are found when the mass-equivalent diameter of rBC core is larger than $\sim 90$ nm.

8. Line 153 frequency is the wrong word.

Reply: We have changed the word 'frequency' to 'frequency distribution' in our revised manuscript (Line 234).

9. Line 185 define K.

Reply: Have defined in the revision (Line 405).

10. Line 255 discuss the Diffusion-Limited Cluster-Cluster Aggregation mechanism (see Sorensen citation above)

[Figure]

Reply: We have read the review literature (Sorensen, 2011) carefully and found that the mass-mobility scaling exponents measured using the tandem techniques (e.g., DMA-APM, DMA-SP2) were sometime erroneously equivalent to the virtual fractal dimension of BC aggregates. The same mistake occurs in our current study. Corrections to this mistake have been performed in the revision (e.g., Lines 87–107, 393–396).

11. Line 261 organics do not 'fill in the gap', they cause restructuring

Reply: Corrected in the revision (Lines 432–434)

12. Line 357 the drag force, or uncertainty? Figure 6 needs error bars. Counting statistics will be poorer at the extreme sizes, possibly causing the observed trend.

Reply: We have rewritten Section 4.3 as shown in Lines 601–613 in the revision. A more reasonable interception is presented for the hiatus in the increase trend of dynamic shape factor.

References Gao, R. S., Schwarz, J. P., Kelly, K. K., Fahey, D. W., Watts, L. A., Thompson, T. L., Spackman, J. R., Slowik, J. G., Cross, E. S., Han, J.-H., Davidovits, P., Onasch, T. B., and Worsnop, D. R.: A novel method for estimating light-scattering properties of soot aerosols using a modified single-particle soot photometer, Aerosol Sci. Technol., 41, 125–135, doi:10.1080/02786820601118398, 2007.

Gysel, M., Laborde, M., Olfert, J. S., Subramanian, R., and Gröhn, A. J.: Effective density of Aquadag and fullerene soot black carbon reference materials used for SP2 calibration, Atmos. Meas. Tech., 4, 2851–2858, doi:10.5194/amt-4-2851-2011, 2011.

Gysel, M., Laborde, M., Mensah, A. A., Corbin, J. C., Keller, A., Kim, J., Petzold, A., and Sierau, B.: Technical Note: The single particle soot photometer fails to reliably detect PALAS soot nanoparticles, Atmos. Meas. Tech., 5, 3099–3107, doi: 10.5194/amt-5-3099-2012, 2012.

Laborde, M., Mertes, P., Zieger, P., Dommen, J., Balternsperger, U., and Gysel, M.: Sensitivity of the Single Particle Soot Photometer to different black carbon types, Atmos. Meas. Tech., 5, 1031–1043, doi:10.5194/amt-5-1031-2012, 2012.

Park, K., Cao, F., Kittelson, D. B., and McMurry, P. H.: Relationship between particle mass and mobility for diesel exhaust particles, Environ. Sci. Technol., 37, 577–583, doi:10.1021/es025960v, 2003.

Raatikainen, T., Brus, D., Hooda, R. K., Hyvärinen, A.-P., Asmi, E., Sharma, V. P., Arola, A., and Lihavainen, H.: Size-selected black carbon mass distributions and mixing state in polluted and clean environments of northern India, Atmos. Chem. Phys., 17, 371–383, doi:10.5194/acp-17-371-2017, 2017.

Rissler, J., Nordin, E. Z., Eriksson, A. C., Nilsson, P. T., Frosch, M., Sporre, M. K., Wierzbicka, A., Svenningsson, B., Löndahl, J., Messing, M. E., Sjogren, S., Hemmingsen, J. G., Loft, S., Pagels, J. H., and Swietlicki, E.: Effective density and mixing state of aerosol particles in a near-traffic urban environment, Environ. Sci. Technol., 48, 6300–6308, doi:10.1021/es5000353, 2014.

Sorensen, C. M.: The mobility of fractal agregates: A review, Aerosol Sci. Technol., 45, 765–779, doi:10.1080/02786826.2011.560909, 2011.

Wang, Y. Y., Liu, F. S., He, C. L., Bi, L., Cheng, T. H., Wang, Z. L., Zhang, H., Zhang, X. Y., Shi, Z. B., and Li, W. J.: Fractal dimensions and mixing structrures of soot particles during atmospheric processing, Environ. Sci. Technol. Lett., 4, 487–493, doi:10.1021/acs.estlett.7b00418, 2017.

Zhang, Y. X., Zhang, Q., Cheng, Y. F., Su, H., Kecorius, S., Wang, Z. B., Wu, Z. J., Hu, M., Zhu, T., Wiedensohler, A., and He, K. B.: Measuring the morphology and density of internally mixed black carbon with SP2 and VTDMA: new insight into the absorption enhancement of black carbon in the atmosphere, Atmos. Meas. Tech., 9, 1833–1843, doi:10.5194/amt-9-1833-2016, 2016.

Please also note the supplement to this comment:

https://www.atmos-meas-tech-discuss.net/amt-2018-408/amt-2018-408-AC1-supplement.pdf

[Figure]

[Figure]

**Fig. 1.** Frequency distributions of incandescence peak height detected by the SP2 for size-selected Aquadag particles at the prescribed mobility sizes

**Fig. 2.** Normalized number size distributions of the rBC core in extBC (red) and thickly coated BC particles (magenta)

**Supplement:**

1 *Supplement of*

**A study of the morphology and effective density of externally mixed black carbon aerosols in ambient air using a size-resolved single-particle soot photometer (SP2)**

Yunfei Wu et al.

*Correspondence to*: Yunfei Wu (wuyf@mail.iap.ac.cn) and Renjian Zhang (zrj@mail.iap.ac.cn)

8    Contents

29

[Figure]

30

**Fig. S1.** Temporal variation of the total number concentration of the scattering and incandescence particles detected by the SP2 during a size selection comprising of one short cycle with a duration of 18 s for each size and one long cycle with a duration of 36 s. The local peak at the beginning of each cycle was previously identified to correct the time difference between the DMA size selection and the SP2 measurement.

36

37

[Figure]

38

**Fig. S2.** Frequency distribution of the lag times between the incandescence and scattering peak

locations determined using the SP2. A bimodal distribution is found, with the minimum at ~2 μs.

The BC-containing particles with a lag time greater than 2 μs were considered to be thickly coated.

Otherwise, the BC-containing particles were non/thinly coated.

43

[Figure]

44

**Fig. S3.** The calibration curves for the relationship between the SP2-measured incandescence peak
height (low gain) and the refractory BC (rBC) mass. The Aquadag particles were used as the
representative material of rBC in the calibration. The Aquadag particles with different mobility
sizes ($d_{mob}$=100, 125, 150, 175, 200, 250, 300, 350, 400, 450 nm, respectively) were manually
selected by a DMA. The masses of the size-selected Aquadag particles were calculated based on
the effective densities ($\rho_{eff}$) according to the density-mobility relationship provided in Gysel et al.
(2011). The manual calibrations were performed both before (on 23 Jan. 2018) and after the
campaign (on 10 Feb. 2018). The red up-triangles represent the relationship between the mass of
Aquadag and the incandescence peak height during the calibration before campaign. The blue
down-triangles represent the calibration after the campaign. It is clear that the SP2 incandescence
peak height is proportional to the mass of Aquadag, inferring the linear regression functions as
y=0.003048x+0.1058 ($R^2$=0.9991) for the calibration before campaign and y=0.002972x+0.1713
($R^2$=0.9992) for the calibration after campaign. In the functions, y represents the mass of Aquadag,
and x represents the incandescence peak height. To examine the detection efficiency of SP2, the
size-resolved Aquadag particles were also automatically selected by the DMA in the $d_{mob}$ range of
140–750 nm and then measured using the SP2 and CPC after the campaign. The
mass-incandescence relationship derived from this measurement was very close to the manual
calibrations. The linear regression function is y=0.003012x+0.1097 ($R^2$=0.9999). The similar rBC
mass calibration curves indicate that the SP2 operated very stably throughout this campaign.
Considering that the incandescence signal peaks of Aquadag were ~25% higher than those of the
ambient BC particles, the slopes of the calibration equations should multiply by a factor of ~1.33
when converting the measured incandescence signal peaks to the rBC masses for ambient
particles.

[Figure]

68

69 **Fig. S4.** Detection efficiency of the SP2 as a function of (a) the mobility diameter of Aquadag and
70 (b) the mass-equivalent diameter of rBC. The detection efficiency was calculated as the ratio of
71 the number concentration of particles detected by the SP2 to that parallel measured by the CPC at
72 prescribed mobility sizes. Since the Aquadag particles were generated by an aerosol generator,
73 pure scattering particles should not exist. The particles having detectable incandescence signals
74 were recorded by the SP2. It is clear that adequate detection efficiencies (>~90%) for the SP2
75 were observed when the $d_{mob}$ of Aquadag was larger than 125 nm. Correspondingly, the
76 mass-equivalent diameter of rBC calculated using the $d_{mob}$ and $\rho_{eff}$ of Aquadag should be larger
77 than ~90 nm to provide adequate detection efficiencies. The relationship between the detection
78 efficiency of the SP2 and the rBC mass can be characterized using an exponential function, as
79 shown in Fig. S4b.

80

[Figure]

81

**Fig. S5.** Frequency distribution of the $d_{me}$ of $ext$BC at 16 prescribed mobility sizes (in the 140–750 nm $d_{mob}$ range) analyzed in the current study. Three thresholds of delay time between the incandescence signal peak and the scattering peak detected by the SP2 were employed to discriminate the $ext$BC particles, e.g., 2.0 µs, 1.2 µs and 0.4 µs. The frequency distributions corresponding to the thresholds are shown in red, green and blue colors, respectively. Reducing the delay time threshold results in a decrease in the data volume used in the statistics, while it seems to have few effects on the peak locations of the distributions. The major peak of the frequency distribution at smaller mobility sizes with a $d_{mob}<140$ nm is not clear due to the lower limit of the SP2. Thus, these smaller sizes were not discussed in the current study.

91

[Figure]

92

**Fig. S6.** The same as Fig. S5, but for the normalized number size distribution (d$N$/dlog$D_c$). The quantity $D_c$ indicates the mass-equivalent diameter of the rBC core, the same as $d_{me}$ defined above. Compared to Fig. S5, these graphs more clearly show that reducing the time-delay threshold has few effects on the peak $d_{me}$. The main changes occur at the right tail of the distributions at each mobility size. Reducing the time-delay threshold results in a significant reduction of particles at the right tail of the number size distribution. These particles are likely to be the thinly or even moderately coated BC particles, which were also recognized as *ext*BC using the time-delay approach. Since the peak $d_{me}$ values are needed in the current study, and are considered the typical $d_{me}$ values for the prescribed $d_{mob}$, these thinly or even moderately coated BC particles should have few effects on our results presented in this study. The peak $d_{me}$ value is identified as the mode value of a lognormal function that is fitted to the major peak of size distribution at each mobility.

105

[Figure]

106

107 **Fig. S7.** Time series of hourly PM$_{2.5}$ mass concentrations measured at the same site during the
108 experimental period. The dashed line represents the mean PM$_{2.5}$ mass concentration with a value
109 of 23.0±26.7 μg m$^{-3}$. The red and blue shaded regions represent a polluted episode (mean
110 PM$_{2.5}$=72.1±23.1 μg m$^{-3}$) and a consecutively clean period (mean PM$_{2.5}$=8.9±2.7 μg m$^{-3}$),
111 respectively, in which the mass-mobility relationships of *ext*BC are compared.
112

[Figure]

113

**Fig. S8.** The mass of *ext*BC particles as a function of the mobility diameter in the range of 140–750 nm (black circles) in the polluted (red up-triangles) and clean (blue down-triangles) episodes. The power-law functions are fitted to the mass-mobility relationships.

117

[Figure]

118

**Fig. S9.** The number fraction of *ext*BC as a function of the mobility diameter ($d_{mob}$) in the polluted
episode (red up-triangles) and clean period (blue down-triangles). The fraction of *ext*BC is roughly
calculated as the ratio of the *ext*BC number concentration to the sum of *ext*BC and internally
mixed BC (*int*BC) particles at each $d_{mob}$. The effect of multicharged particles is not eliminated.
However, the multicharged particles should result in a similar effect on *ext*BC and *int*BC at a
given mobility. Thus, there should be few effects on the number fraction of *ext*BC. A detailed
analysis of the mixing states of size-resolved BC particles will be performed in our future studies.

126

127  **Table S1.** The typical mass-equivalent diameters ($d_{me}$) and corresponding masses of *ext*BC at
128  different mobility sizes ($d_{mob}$) selected by the DMA in the whole campaign. Three delay-time
129  thresholds of 2.0, 1.2, and 0.4 µs are employed to discriminate *ext*BC.

| $d_{mob}$ (nm) | $d_{me}$ (nm) | | | mass (fg) | | |
|---|---|---|---|---|---|---|
| | $t_{lag} < 2.0$ µs | $t_{lag} < 1.2$ µs | $t_{lag} < 0.4$ µs | $t_{lag} < 2.0$ µs | $t_{lag} < 1.2$ µs | $t_{lag} < 0.4$ µs |
| 140 | 88.8 | 88.4 | 86.7 | 0.66 | 0.65 | 0.61 |
| 160 | 97.5 | 97.6 | 96.2 | 0.87 | 0.88 | 0.84 |
| 180 | 106.2 | 106.6 | 105.0 | 1.13 | 1.14 | 1.09 |
| 200 | 115.6 | 116.2 | 114.2 | 1.46 | 1.48 | 1.40 |
| 225 | 127.9 | 128.8 | 125.8 | 1.97 | 2.01 | 1.88 |
| 250 | 140.5 | 141.6 | 138.1 | 2.62 | 2.67 | 2.48 |
| 280 | 155.8 | 156.9 | 152.9 | 3.56 | 3.64 | 3.37 |
| 315 | 172.6 | 173.6 | 169.4 | 4.85 | 4.93 | 4.58 |
| 350 | 188.2 | 189.2 | 184.7 | 6.28 | 6.38 | 5.94 |
| 400 | 207.4 | 208.6 | 204.8 | 8.41 | 8.55 | 8.10 |
| 450 | 226.4 | 228.0 | 224.1 | 10.94 | 11.17 | 10.61 |
| 500 | 243.8 | 245.4 | 241.2 | 13.65 | 13.92 | 13.23 |
| 560 | 262.6 | 265.4 | 262.0 | 17.06 | 17.62 | 16.96 |
| 630 | 283.2 | 286.1 | 285.3 | 21.42 | 22.07 | 21.88 |
| 700 | 305.1 | 307.9 | 309.8 | 26.76 | 27.52 | 28.03 |
| 750 | 319.6 | 322.3 | 323.4 | 30.76 | 31.57 | 31.87 |

130

---

## Author Comment (AC2) · 5 Mar 2019

**General comments**

The authors use a tandem DMA-SP2 system to measure the mass-mobility exponent, effective density, and shape factor of ambient BC particles in Beijing, China. Despite the claim to novelty made in the title similar combined DMA-SP2 measurements have already been discussed and reported in the literature, which the authors have failed to discuss. In addition, there are significant shortcomings in the manuscript itself, ranging from inadequate description of the experimental details and results to data quality issues. In my view the manuscript is at the 'early draft' stage rather than the 'under review' stage. I believe major revisions are required before the manuscript can be considered for publication in AMT. Most importantly, the authors need to better demonstrate why this study is an original contribution to the literature on the properties of ambient BC aerosols, and why their measurements should be considered artifact-free and trustworthy.

We appreciate the criticisms from the reviewer and appreciate the several useful comments which helped us improve the paper. A huge modification has performed to our manuscript by referring more useful literature, reanalyzing the experimental data and improving the English writing. We hope the reviewer will see our efforts to improve the quality of our manuscript.

I see the following major issues:

The study is not placed in appropriate context through citation of relevant literature. In the title and elsewhere (e.g. L101) the authors claim that this is a 'novel' system or measuring the morphology and effective density of black carbon particles. This is incorrect. Tandem DMA-SP2 measurements have been discussed and/or performed in a number of different studies (e.g. Gysel et al., 2011; Raatikainen et al., 2017; Zhang et al., 2016). A handful of references are cited for more general mass-mobility measurements (e.g. using an APM rather than an SP2). But the literature on this topic is more extensive than this small selection of studies would suggest, including a review on the mobility of fractal aggregates by Sorensen (2011). I suggest the authors read this review and the references therein and put more effort into placing their measurements into the context of these previous works. In particular, the authors need to demonstrate what is the original contribution of this work.

Reply: Thanks for the constructive comments. Indeed, a number of previous studies have used the tandem DMA-SP2 system to measure the physical and/or optical proper-
ties of BC particles. A few related references have been cited in the initial manuscript (e.g., Gysel et al., 2011; Zhang et al., 2016), although a few other references were missed (e.g., Raatikainen et al., 2017). Citation of these related references has been performed in our revised manuscript. As mentioned in the introduction of the revision (Lines 129–141), the mass-mobility relationship, from which the morphology and effective density of extBC aggregates are obtained, were seldom studied in previous DMA-SP2 measurements, especially in the ambient atmosphere. The morphology and effective density of internally mixed BC (intBC) particles in the atmosphere in North China Plain were studied by using the VTDMA-SP2 measurement (Zhang et al., 2016).

We should not miss the valuable review literature (Sorensen, 2011) in our initial manuscript. We have read the literature carefully according to the reviewer's comment and found that the 'mass-mobility scaling exponent' was erroneously considered the virtual 'fractal dimension' even they have a corresponding relationship and both reflect the morphology of BC aggregates. Similar mistakes also existed in previous studies on mass-mobility relationship (e.g., Park et al., 2003). Corrections can be found in Lines 87-107, 393-396 in the revision. By carefully referring to this review article, we have improved our understanding in the subject of the morphology of BC aggregates. Then we further condense our research purpose as shown in the rewritten introduction of the revision. As presented there, although the morphology and effective density of BC aggregates have been widely studies using the TEM analysis and/or tandem techniques (e.g., DMA-APM), such studies were mainly conducted in the laboratory and/or in the source environments (e.g., in the tunnel) where the freshly emitted soot aggregates were predominant. The SP2 provides an advantage to distinguish the extBC aggregates from the thickly coated BC particles at the same time to the rBC mass determination. Thus, the DMA-SP2 can be deployed to different atmospheric environments to investigate the mass-mobility relationship of ambient extBC. Another advantage of the DMA-SP2 tandem system is that it can investigate the mass-mobility relationship of BC aggregates in a relatively larger mobility range (e.g., in the 350-750 nm mobility diameter range) due to the high sensitivity and accuracy of the SP2 in the rBC mass
determination in this range, allowing to examine the applicability of the mass-mobility relationship obtained from previous studies and the consistence of the mass-mobility relationship determined using different tandem techniques. Previous DMA-APM measurements seldom concerned the BC aggregates in the atmosphere with mobility diameter (dmob) larger than 350 nm due to the larger measurement uncertainty in the larger particle sizes. Thus, our study should have its practical and scientific meaning, even the DMA-SP2 tandem technique is not novel. We have removed the word 'novel' in the title of our manuscript according to the reviewer's comment.

The quality of the writing is not at a suitable level for scientific publication. There are many English grammar issues - too many to list in a scientific review. Beyond this, the language is frequently too vague. To take just the first example I come across the authors state that BC can lead to 'Earth warming' on L39. I believe that authors mean 'warming of the Earth's atmosphere' or similar. There are many more examples of such lazy language throughout the manuscript.

Reply: Thanks for the professional comments. Actually, the writing of our manuscript had been polished by native English speakers before it was submitted to the journal for review. They might have made an improvement in the English grammar and writing. A number of more professional contents (e.g., the example posed by the reviewer) are still needed to improve. We checked carefully throughout our manuscript to make corrections to these similar mistakes. The revised manuscript has been further polished by a more professional agency before resubmitted for review. We hope you can see the improvement in the English writing of our revised manuscript. Besides, we also have made efforts to improve the scientific quality of this article by rewriting a large fraction of the manuscript.

The study design and experimental and analytical details are inadequately described, which makes it difficult to judge their suitability. For example from what I can gather, the results presented in Figs. 2 to 6 are only for rBC-containing particles that displayed a delay time of less than 2us (defined as 'extBC'). This needs to be clarified and stated
more explicitly (another example of lazy language). The suitability of using delay time to distinguish between externally and internally mixed BC particles then requires further discussion. It is not as simplistic as the authors make it out to be. The authors allude to the fact that thinly-coated BC will exhibit low SP2 delay times on L280. This is also true for moderately-coated BC (e.g. with BC volume fractions as low as 30%), which would certainly not be classified as 'externally mixed' (ExtBC). A better approach for classifying BC mixing state with the SP2 is the quantitative LEO-fit approach (Gao et al., 2007; Laborde et al., 2012). The authors need to at least discuss this more sophisticated method, the reasons why they chose to use the more simplistic delay time approach, and the consequences of this decision.

Reply: The manuscript is written to discuss the physical properties (mainly the massmobility relationship) of externally mixed BC (extBC) particles, although the experiment was designed for a range of research aims. For example, the experiment was originally designed to also study the mixing states (e.g., the fraction of thickly-coated BC particles and the coating thickness) and optical absorption properties (e.g., mass absorption efficiency) of size-selected BC particles, and the number/mass size distribution of BC particles in the atmosphere, similar to the experimental setup presented in Raatikainen et al. (2017). Two micro-aethalometers used to determine the aerosol absorption coefficients were also connected to the DMA and measured parallel with the SP2 (as shown in Fig.1 in the manuscript). The data analysis is still in progress. Several difficult but crucial issues should be deal with in data processing, e.g., the effects of multicharged particles. At the current stage, we present our first study on the mass-mobility relationship of the extBC particles, although the effects of multicharged particles are also needed to take into consideration. A number of the technical details associated with this experiment are described in current manuscript. Further studies mentioned above are also in preparation. Similar statements are also presented in the revised Section 2 (Lines 201-206).

In this study, the extBC particles are discriminated according to the delay time be-
tween peaks of the incandescence signal and the scattering signal measured by the SP2. Only the BC-containing particles with the delay times less than the criterion are recognized as the extBC particles. The remaining BC-containing particles were considered thickly coated by non-refractory components. The criterion is determined from the frequency distribution of delay time of individual particles. As shown in Fig. S1, the frequency distribution of delay time exhibits a significantly bimodal pattern. The delay time corresponding to the minimum of the frequency distribution between the bimodal peaks is chosen as the criterion. In this study, the criterion of the time delay is selected as 2.0  $\mu$ s. Indeed, a fraction of thinly coated BC particles might be also recognized as the extBC based on the time-delay approach. A few moderately coated BC particles (e.g. with BC volume fractions as low as 30%) might also have delay times less than the criterion as mentioned by the review (Laborde et al., 2012), which might be mistakenly recognized as the extBC. To examine the stability and reliability of our results derived by using the time delay approach, we strengthen the discrimination of mixing state by decreasing the criterions of delay time from 2  $\mu$ s to 1.2  $\mu$ s and 0.4  $\mu$ s. Reducing the delay times to less than 0.4  $\mu$ s means that the incandescence signal has the same peak location to the scattering signal because the minimum unit for the signal record is 0.4  $\mu$ s. The effects of thinly and even moderately coated BC particles are expected to decrease when the delay time threshold decreased from 2.0  $\mu$ s to 1.2  $\mu$ s, and then to 0.4  $\mu$ s. However, as shown in Fig. S5 (the frequency distributions of mass-equivalent diameter of rBC core (dme) of the extBC particles, which is calculated from the measured rBC mass by assuming the rBC core is a void-free sphere with 1.8 g cm-3 density, at the prescribed mobility size ranges) and Fig. S6 (the normalized number size distribution of rBC core of the extBC particles) in the revised supplemental file, the decrease in the delay time threshold has insignificant impacts on the peak location of frequency or number size distribution, i.e., the typical dme of extBC at each prescribed mobility size, although the data volume used to calculate the distribution reduced significantly. As also shown in Table S1, the discrepancy in the fitted peak dme values resulting from the different delay time thresholds don't excess 3% in the
current study. It means the differences in fitting peak values of rBC mass don't excess 10%, which is considered as the uncertainty in the determined rBC mass at each mobility size due to the time-delay method employed to distinguish the extBC particles from the thickly coated BC particles. The thinly coated and even moderately coated BC particles mistakenly considered as the extBC discriminated by using the time-delay method appear to influence the shape of the size distribution. These particles mixed with multicharged extBC particles resulting in an increase in the size distribution at its right tail. At the current stage, we are unable to separate which particles presented in the right tail are thinly/moderately coated and which are multicharged. However, this issue should affect little on the results presented in our current study. Similar statements are also presented in the revised Section 3.2 (Lines 344–379 in the revision).

Actually, the thickness of a BC-containing particle can be retrieved using a specific approach, i.e., lead-edge-only (LEO) fitting (Gao et al., 2007). The method can be used to estimate the optical size of the particle using the scattering signal detected by SP2 at its initial stage (e.g., the first 5%) by assuming certain parameters, including the refraction indices of coating material and the rBC core. By comparing with the mass-equivalent diameter of the rBC core determined by using the incandescence signal, the thickness of the coating can be estimated. However, this approach needs the scattering signal measured by the split channel (with a two-element APD) of the SP2, which is used to determine the center location of the Gaussian-distributed laser beam. Unfortunately, this channel of our SP2 was out of work during the experiment. We failed to use this method to estimate the coating thickness of the BC-containing particle in the present study. However, the 'coating thickness' is not a crucial quantity in our current study because only the extBC particles without coating or with thin coating were concerned. The 'coating thickness' can be used to verify the results currently presented. Similar statements can also be found in the revised Section 2.3 (Lines 248-267 in the revision)

We are also preparing a new experiment which is similar to the presented one by
employing a SP2 with all detect channels well performed. Moreover, because a number of assumptions are employed in the LEO fitting and the subsequent Mie calculation, large uncertainties are also exist in the retrieved coating thickness. For instant, the size of the rBC core (dme) is determined from the rBC mass (calculated according to the linear relationship between incandescence single peak and rBC mass calibrated before and after the experiment) by assuming it is a void-free sphere with 1.8 g cm-3 density. If the BC particle is not void-free and has a low density (Zhang et al., 2016), the size of the rBC core might be underestimated. In addition, the optical size of the particle largely depends on the refraction indices of the coating materials and the rBC core, which are usually known for a specific site. Generally, the coating thickness is not a crucial quantity in our current study of the morphology and density of uncoated extBC aggregates. It can provide a validation of our discrimination of extBC but should have little influence on our final analysis and discussions presented in current study.

The material as presented gives cause to question the quality of the measurement data, but insufficient details are provided to fully make this assessment. Specific points are highlighted below in relation to Fig. 2, which contains features that require explanation (absence of clearly defined peaks for multiply charged particles, presence of peaks with rBC mass approaching 0). In addition for reasons that are not yet satisfactorily explained, the effective density results presented in Fig. 5 are systematically lower than previous measurements (both the previous studies already cited by the authors as well as previous DMA-SP2 measurements that were also conducted in Beijing but are not yet discussed; Zhang et al., 2016). Given these issues, the authors should include further data and explanations to build confidence in their results and to confirm that the measurements are artifact-free. For example, were PSL spheres or other monodisperse particles (e.g. aquadag for the SP2 measurements) used to confirm that the DMA was operating correctly? What quality checks were performed to ensure the SP2 was operating correctly? (E.g. laser and detector block properly aligned, laser power levels, flow rate checks, comparison of calibration curves with previous calibration curves of the instrument).
Reply: Thanks for the comments, we have increased the description in the detail of the experiment and data processing in the revision. Meanwhile, we used the number size distribution of rBC core of the extBC particles in the revision instead of the initial frequency distribution of rBC mass of the extBC particles. The normalized number size distribution is a more practical quantity in the scientific research and has lower artificial uncertainties than the frequency distribution which depends on the bin-size used to calculate the frequency. We have completely reanalyzed our dataset and present more reliable results in the revision. Discussions on the measurement uncertainty are also presented. Specific responses to the review's queries are presented point-to-point below.

1. Peaks with rBC mass approaching 0 presented in the frequency distribution of rBC mass are likely attributed to the measurement noises of the SP2. Explicit discussions are also presented in the response to the No.18 specific comment. The smaller rBC masses approaching 0 were also observed in the SP2's incandescence calibration using size-selected Aguadag particles (Fig. R1). As shown in Table R1, the number contribution of measurement noises gradually increases from 1% at 140 nm dmob to 37% at 750 nm dmob. The noises should be resulted from the SP2 measurement only and not presented in the CPC measurement. As shown in Fig. R2, if the noises are not eliminated, the counting efficiency of SP2 (calculated as the ratio of number concentration measured by the SP2 to that counted by the CPC at a given mobility) shows an increase trend with increasing dmob. While after correction to the effect of these noises, the counting efficiency of SP2 is stably close to 100% in the 125-750 nm dmob range. We have used the normalized number size distribution of rBC core of the extBC particles instead of the initial mass frequency distribution. The effect of these small rBC masses on the number size distribution is significantly reduced due to data processing.

As also shown in Fig. R1 and Table R1, the doubly charged particles shown a discernable minor peak in the frequency distribution of the incandescent peak height in the
calibration using Aquadag particles. Since the Aquadag particles generated from the aerosol generator can be considered as the bare BC, the mobility of Aqudag particles should be only related to their masses and morphologies, resulting in the clear peak of multicharged particles. However, in the ambient atmosphere, it is more complex even for the extBC particles. As mentioned above, a fraction of thinly coated particles is also recognized as extBC based on the time-delay discrimination. These particles coexisted with the multicharged particles result in a combined effect on the size distribution of rBC core of the extBC particles at a given mobility. They mainly result in an increase in the size distribution at its right tail and have few effects on the determined typical dme of extBC. Similar statements can be found in Lines 344–379 in the revision. Anyway, based on the clear distinguishable minor peak presented in the frequency distribution of the incandescence peak height for the multicharged bare BC particles (Aquadag particles) and the adequate detection efficiency of SP2, we propose the DMA-SP2 system operated normally in this experiment.

2. The observation data have been reanalyzed. The effective densities of extBC obtained from the recalculated mass-mobility relationship at the prescribed mobility sizes are slightly lower than those obtained from the previous DMA-APM measurements. The deviations are general in the measurement uncertainty of DMA-SP2 system (~20%). The slightly lower densities of extBC aggregates are also likely due to the differences in the mass determination between APM and SP2. The APM measures the mass of whole BC aggregate, which might be composed of a fraction volatile or semivolatile materials in addition to the primary BC spherules. While the mass of rBC is determined by using the SP2. Thus, a slightly lower mass of extBC aggregates can be expected at a given mobility size. Meanwhile, the volatile and/or semivolatile materials might also result in a more compact structure of extBC aggregate due to the reconstruction effect by these materials thinly coated outside the primary BC particles. The extBC mass determined as the peak rBC mass of the size distribuition of rBC core of the extBC particles might mostly correspond to the uncoated extBC which are less influenced by the volatile and/or semivolatile materials. It also results in the slightly

AMTD
lower densities obtained from the DMA-SP2 measurement than those from previous DMA-APM measurements. Anyway, the densities derived from the reanalyzed dataset are generally comparable to previous values for diesel soot particles in the laboratory and in the source environments such as in the tunnel.

The effective densities of thickly coated (internally mixed) BC (In-BC for short in the reference) particles were presented in Zhang et al. (2016). Due to the significant reconstruction by the nonrefractory components thickly coated outside the primary BC particles, these BC particles became more compact resulting in obviously higher effective densities than the uncoated BC aggregates presented in current study.

3. Before this experiment, the laser alignment was performed by the operator according to the manual provided by manufacturer step-by-step. Besides the incandescence calibration using Aquadag particles, the scattering calibration using PSL with a certain diameter (269 nm) were also carried out before and after the campaign. A slight decrease in the measured scattering peak height (low-gain) for 269 nm PSL was observed after the campaign compared to that before the campaign (from 3576 to 3555 a.d.). It indicates the laser of the SP2 was stable throughout the campaign. Since the split channel was out of work in the experiment, the absolute value of the scattering peak height for 269 nm PSL appears to be useless in this study, which is mainly used in the retrieval of particle optical size by using the LEO fitting method. In the scattering calibration of the SP2, we also delivered the 269 nm PSL to pass through the DMA before to be measured by the SP2. We examined the SP2 recorded particle number concentrations by adjusting the mobility diameters of particles through the DMA. Because the PSL particles are general spheres, their mobility diameters are equal to their geometric diameter. Thus, we found that the recorded number concentration showed a significant peak when the mobility was set to  $\sim$ 269 nm for the 269 nm PSL in the calibration. It indicates the DMA was operated correctly although the number concentration was not recorded in the computeãAC

Generally, the DMA-SP2 system was operating correctly in the experiment.
Fig. R1 Frequency distributions of incandescence peak height detected by SP2 for size-selected Aquadag particles at the prescribed mobility sizes. The Gaussian fitting is performed for the major peak of each distribution to obtain the typical incandescence peak height at the corresponding mobility size.

Fig. R2 Detection efficiency of the SP2 to Aquadag particles with different mobility diameter in the range of 50–750 nm selected by the DMA. The left panel shows the detect efficiency of SP2 including the effect of measurement noises and the right is noise-corrected detection efficiency.

Table R1 Number fractions of single charged particles, detection noises, and multiply charged particles at different mobility sizes in the measurement for Aquadag particles using the DMA-SP2 tandem system. As shown in Fig. R1, the high frequencies of incandescence signal peak height close to 0 are considered to be attributed to the detection noises. The major peak of the frequency distribution is attributed to single charged Aquadag particles and the minor peak at the right side of the major one is considered to be resulted from double charged Aquadag particles.

No uncertainty estimates are provided for the measurements and main results. This is important as the SP2 counting statistics and therefore measurement errors will be sizedependent. How sensitive are the reported quantities (e.g. Df) to these measurement uncertainties?

Reply: Discussions on the uncertainty in the determination of extBC mass have been performed in the revised manuscript. The uncertainty in the typical extBC mass at each mobility is arisen mainly from two aspects. One is the uncertainty in the rBC mass measured by the SP2. It is mainly related to the uncertainty in SP2 incandes-cence measurement and the calibration factor that is determined from the rBC mass-incandescence relationship using the standard soot particles (Aquadag particles in this study). Approximately 3% variation in the calibration factor is estimated throughout the experiment. Both considering this deviation and the uncertainty in the measured incan-

AMTD
descence signal and calibration material, we estimate the uncertainty in the measured rBC mass to be ~10%. The other uncertainty is arisen from the time-delay approach used to discriminate the extBC particles. By examining the effect of delay time threshold on the peak dme, the associated uncertainty is estimated to not exceed 10%. Thus, the total uncertainty in the typical extBC mass used in the further mass-mobility relationship analysis is ~20%. Similar discussions can also be found in Lines 277–283, 371–376 in the revision.

The counting efficiency of SP2 is calculated as the ratio of particle number concentration measured by the SP2 and that measured by the CPC at a given mobility selected by the DMA. The counting efficiency was examined using the Aquadag particles after the campaign. As shown in Fig. S4, the SP2 has adequate counting efficiencies (>90%) for Aquadag particles with mobility diameter (dmob) larger than 125 nm. The counting efficiency is stably close to 100% in the dmob range of 125–750 nm. We converted the dmob to mass-equivalent diameter (dme) of rBC according to the size-dependent effective densities of Aquadag particles (Gysel et al., 2011). The SP2 has adequate counting efficiencies (>90%) in the  $\sim$ 90–420 nm dme range. Thus, the mass-mobility relationship of the extBC particles in 140–750 nm dmob range presented in the current study should be reliable and has high confidence.

We propose the uncertainties in the determined extBC mass are similar at different mobility. It means if a lower (or higher) extBC mass is expected due to the uncertainty at a given mobility, the lower (or higher) extBC mass at other mobility sizes should also be expected. Although we don't discuss the sensitivity of mass-mobility scaling exponent to the measurement uncertainty in the manuscript, its sensitivity to the fitted size range is presented in the revision (Lines 450–457). Meanwhile, the deviation in the mass-mobility relationship in a polluted episode compared to that in a clean period also investigated (Lines 471–499 in the revision).

Specific comments
L69: 'Representativeness' instead of 'representation'

Reply: Thanks for the careful review. We have rewritten the introduction of our manuscript.

L80 - 82: The meaning of these sentences is not clear, rewording required

Reply: The initial purpose of the sentences is to express how the APM determines the mass of particles at a given mobility selected by the DMA. We have rewritten these sentences as shown in Lines 114–118 in the revision.

L92: Schwarz misspelt.

Reply: Have corrected in the revision (Line 129).

L98: Statement needs qualification. This is not true when measuring pure BC particles

Reply: We have rewritten the whole paragraph in the revision to highlight the practical significant and advantage of our study on morphology and effective density of ambient extBC in the atmosphere of urban Beijing using the DMA-SP2 tandem measurement (Lines 124–172).

L109: Please include information about the neutralizer that was used upstream of the DMA. This is especially required when considering the potential impacts of multiply charged particles as discussed in Section 3.2.

Reply: A Kr neutralizer (model 3087, TSI Inc.) was utilized upstream of the DMA to charge the particles entering into the system. A simple description of the neutralizer has been added in the revised manuscript (Lines 185–186).

L135: Please include the length of the nafion dryer and the source of the dry sheath air.

Reply: A model MD-700-12F-3 (Perma Pure LLC, Toms River, NJ, USA) nation dryer with length of 12 inch was used. The total sample flow rate passing though the nation
dryer was  ${\sim}0.8$  L/m. The dry sheath air was drawn by a Vacuum pump (KNF) opposite to the sample flow direction. We added these information in our revised manuscript (Lines 187–191).

L139: Some discussion is required about SP2 counting efficiency over this size range. It is not always 1, which will affect counting statistics, adding uncertainty to measurements reported at the limits of the range.

Reply: Although 33 mobility diameters was selected in each cycle, only the mobility diameters (dmob) in range of 140–750 nm are analyzed because the size distribution of rBC core of the extBC particles with a dmob smaller than 140 nm cannot exhibit a clear peak, from which the typical mass-equivalent diameter (dme) of extBC is determined.

The detection efficiency of SP2 is shown in Fig. S4 in the revised supplemental file by comparing the number concentration of generated Aquadag particles measured by the SP2 to that by the CPC at each mobility. As shown in Fig. S4, the SP2 has adequate detection efficiencies (>90%) for Aquadag particles with a dmob not smaller than 125 nm, equivalent to a dme larger than ~90 nm. Thus, the detection efficiency of SP2 should not have great effects in the 140–750 nm dmob range we concerned in the current study. Since the peak dme of the size distribution of extBC at each mobility is require, the low detection efficiency of SP2 to small particles should influence little of the results derived from our analysis.

L153: The phrase 'frequency of the time lag' does not make sense. I guess the authors meant frequency distribution or histogram.

Reply: Thanks. We used the 'frequency distribution' instead of 'frequency' in the revision (Line 224).

L156: Am I correct in assuming that the 'extBC' results presented later only include particles that displayed a lag time less than 2 us? If so please state this explicitly.

Reply: Yes, we only analyzed the mass-mobility relationship and effective density of
'extBC' distinguished according to the delay time between the incandescence peak and the scattering peak with lag times less than 2  $\mu$ s. Explicit statements are shown in Lines 230–247 in the revision.

Also including in the revision is the examination of the effect of delay time threshold (decrease from 2  $\mu$ s to 1.2  $\mu$ s and 0.4  $\mu$ s) on the derived mass-mobility relationship of extBC. A decrease in the lag time threshold means a stricter discriminant criteria for the extBC particles using the time-delay method. It is discussed in detail in Section 3.2 of the revision.

L162: More sensitive than what? To Fullerene soot I presume but statements like this need to be explicit, avoiding lazy language.

Reply: The incandescence signal is more sensitive to Aquadag particles than to the Fullerene or diesel exhaust soot particles which are the mainly BC particles in the atmosphere, e.g., the same mass of Aquadag particle results in a high incandescence peak than the Fullerene soot or diesel exhaust soot. Similar corrections can be found in Lines 271–274 in the revision. We have check carefully throughout our manuscript to avoid similar writing errors.

L166: Please include a Figure in the supplementary information of the two measured incandescence calibration curves. Were the incandescence calibrations performed all the way up to 750 nm or were the calibration curves extrapolated? What sort of function was fit to the calibrations curves? Were the calibrations consistent with previous calibration curves measured for this instrument (an important check to make to ensure the SP2 was operating ok)

Reply: The calibration curves before and after the campaign are showed in the revised supplemental file (Fig. S3). In the two manually calibration for the incandescence signal of the SP2, Aquadag particles with the dmob in the range of 100 nm to 450 nm (e.g., 100, 125, 150, 175, 200, 250, 300, 350, 400, 450 nm) was measured by the SP2. As shown in Fig. S3, the incandescence peak heights detected by the SP2 are

AMTD
linearly correlated with the rBC masses which are calculated according to the effective densities of Aquadag particles provided in Gysel et al. (2011). After the campaign, we also used the DMA-SP2 tandem system to measure the generated Aquadag particles automatically. The setup of the DMA-SP2 system was the same as that used to measure the ambient particles. Thus, the relationship between the incandescence peak height and rBC mass obtained from the size-selected Aquadag in the dmob range of 140–750 nm is established. As shown in Fig. S3, the linear relationship between the incandescence peak height and rBC mass is robust in the 100–750 nm dmob range for our SP2. The calibration factors (the slopes of the linear regressions) vary little during this campaign, indicating the good performance of our SP2. The good performance of our SP2 can also be validated by the high detection efficiency (>90%) of our SP2 for BC particles with dme larger than ~90 nm (Fig. S4).

L176: Please also discuss what the prefactor k represents (e.g. Sorensen 2011).

Reply: Since the mass-mobility relationship is studied, the prefactor k here is not consistent to that in the fractal relationship of DCLA aggregates presented in Sorensen (2011). The prefactor k (actually log(k)) is the intercept of the linear regression of extBC mass against its mobility diameter in the logarithm scale. The k value obtained in current study is in the same order of magnitude to the previous values for soot aggregates (Park et al., 2003). Explicit discussion can also be found in the response to the No.19 specific comment. Specific discussion on k will not presented in the current manuscript.

L191: Please provide a reference for why this value of material density was chosen.

Reply: A reference, Taylor et al. (2015), is added in our revised manuscript (Line 411).

L199: The more common phrasing for this section would be 'Data processing'. And it seems to me that Section 2.4 would be a better fit in this section rather than the measurement methodology section above.

AMTD
Reply: Thanks for the constructive comment. We have modified title of Section 3 (Lines 286–287, Line 325) and moved the section 2.4 to 3.3 according to the suggestion.

L220: It seems that this method was not used in this study. Why is it mentioned? Was it used to check the results processed with the first method?

Reply: At the beginning of the experimental setup, we intended to use the correlation method. However, during the data analysis, we found that size distributions of SP2-detected particles were inadequate for the further calculation of the correlation coefficients since the detection efficiency of the SP2 decreases dramatically in the small particle range. Therefore, the local peak method is developed in current study to identify the time difference between the size selection and the SP2 measurement. The correlation method will be used to examine the time difference between the size selection and the AE51/CPC measurements in our future study of the number and mass size distribution of BC. Similar statements are presented in the revision (Lines 317–323).

L232: 'Minimizing the multicharged particles' is too vague. Suggest 'Correcting for the presence of multiply-charged particles' or something similar.

Reply: Thanks. We adopt the suggestion of the reviewer by changing the title of Section 3.2 as 'Determination of the typical masses of extBC at prescribed mobility sizes'. We have also rewritten the whole content of this section to express explicitly how we determine the typical masses of extBC at prescribed mobility sizes. The effect of the delay time threshold chosen for the discrimination of extBC on the determined extBC masses is also discussed in the revision (Lines 326–379).

L240: Please be more specific. Exactly what parameter of the fitted curve was used to represent the mass of singly charged extBC particles?

Reply: In the initial manuscript, the mean value or expectation ( $\mu$ ) of the Guassian distribution f(x)=Aexp(-ãĂŰ(x- $\mu$ )ãĂŮ2/(2 $\sigma$ 2)) was used to represent the mass of singly charged extBC.
We use the number size distribution instead of the rough frequency distribution of rBC core mass of the extBC particles in the revision. The peak dme determined as the mode value of the lognormal function fitted to the major peak of the number size distribution of extBC is considered the typical dme of extBC (singly charged) at a given mobility. We have rewritten the whole content of Section 3.2 in the revision.

Fig. 2: These curves contain features that require discussion to build confidence in the measurements. For example: related to the comment above about L156, do these frequency distributions only contain particles that display time lag less than 2us? If so, why are there sharp increases in the number of particles with rBC mass approaching 0 at diameters greater than 160nm? (With no filtering of the data applied I would assume these are heavily coated particles, which is a reason why I think the filtering process is not strict enough to be able to label these particles as 'extBC'). Why do the multiplycharged particles show up as a very fat tail that does not descend to a frequency of 0 until some point beyond the upper limit of the x-axis? What do these tails in the distribution represent? (E.g. they are in contrast to what is typically seen in SP2 calibrations, when doubly and triply charged particles along the x-axis corresponding to the mass of particles selected by DMA (e.g. under the assumption of spherical particles with the material density of BC) to provide reference points to compare the measured mass distributions to.

Reply: The normalized number size distribution (dN/dlogDc, where Dc is the massequivalent diameter of rBC core by assuming it is a void-free sphere with 1.8 g cm-3 density, N is the number concentration in a given Dc range) instead of the initially rough frequency distribution of extBC mass at each prescribed mobility in the range of 140-750 nm is presented in Fig. S6 in the revised supplemental file. Five normalized number size distributions at 140, 225, 350, 500, 750 nm dmob are selected and also shown in Fig. 2 in the revision. The major peaks are more significantly through such data processing at prescribed mobility sizes. The size distributions of rBC core at
different mobility sizes are generally used in the studies on size-resolved BC particles. For instant, Zhang et al. (2016) measured the morphology and density of internally mixed BC (In-BC for short in the reference) with SP2 and VTDMA. They used the peak diameter of the normalized volume size distribution of the In-BC core to represent the typical diameter of In-BC core at the prescribed mobility diameters selected by the DMA.

We originally considered the sharp increases in the number of particles with very small rBC mass resulted from the detection noises of SP2 for the smaller BC particles. As claimed by the manufacture, the lower BC detection limit is  $\sim 0.3$  fg per particle. Thus, data filter was performed by eliminating the particles with incandescence signal peak values lower than 100. This criterion is determined from the calibration curve by using Aquadag particles (Fig. S3 in the revised supplemental file). However, the distribution concentrated at the smaller masses still existed after the data filter. Initially, we suspected the smaller masses belonged to the thickly-coated BC particles which were not separated from the extBC particles using the time delay method. However, we found that the smaller masses also existed for the bare BC particles (i.e., Aguadag). As shown in Fig. R1 and Table R1, the particles with these smaller masses accounted for only  $1 \sim 9\%$  of the total particles (after data filter by eliminating the signal with incandescence peak value lower than 100) in the range of 140-200 nm in mobility diameter. This proportion gradually increased to  $\sim$ 37% at 700 nm. Thus, these smaller masses should not be resulted only from the 'remnant' BC particles with thick coating. We still attribute these smaller masses to measurement noises.

The Gaussian fitting for the major peak can distinctly distinguish the singly charged particles from those with double or more charges for Aquadag particles (considered as bare BC particles). However, in the case of measurement for ambient particles, it is more complex. First, the time delay approach used in this study can only distinguish the thickly-coated BC particles from those with thinly or without coating. A fraction of thinly- and/or even moderately-coated BC particles cannot be absolutely separated
from the bare BC particles. Thus, a fraction of extBC particles claimed in this study might be thinly-coated ones. We considered the right tail of the distribution was also associate with these thinly-coated BC particles. These thinly-coated BC particles has a relatively more compact structure than the bare BC particles, resulting in larger masses in a given mobility diameter.

L252: Please provide further details for how the fit was performed. Were both the prefactor k and mass-mobility exponent Df allowed to vary freely? The k value seems to be very low in comparison to previous measurements (Sorensen 2011 and references therein), which requires explanation. How is the standard deviation of 0.04 in Df determined? How sensitive is the fitted Df to the size-dependent measurement uncertainties in rBC mass?

Actually, Reply: a linear regression of log(mass) (e.g., against **y**) by minimizing the log(dmob) (e.g., X) chi-square error statistic between the fitted log(mass) (e.g., yfit) and measured ones. The chicalculated  $\sum (y - yfit) 2. According to the mass$ as square error is mobility exponential equations how nin Section 3.3 of the revision, theyinterceptof the linear regression should be log(k), and the slope is Dfm. We have corrected the definition of fractal dimension (Linear regression should be log(k)).

mobility relationship cannot compare to that presented in Sorensen (2011). It seems to be comparable to the value presented in prevent of the solution of th

 $8 \times 10-6$  (Park et al., 2003), close to the value presented in our study. The prefactor of mass-mobility relationship is with few concern in previous studies.

The standard deviation in Dfm is derived as the 1-sigma uncertainty estimates for the slope of linear regression of extBC mass against dmob at the logarithmic scale. The uncertainty in the determination of extBC was extimated to be  $\sim$ 20% at each dmob, including the uncertainty in the rBC mass determined by the SP2 and that raised from the time delay method used to discriminate extBC. Similar discussion can found in our
revised manuscript, e.g, Lines 358-365, 463-469.

Fig. 3: Error bars are required in this figure to indicate the measurement uncertainties.

Reply: The typically mass of extBC was determined from the lognormal fitting. We also show the uncertainties in the revised Fig. 3 and Fig. 4, and discuss the uncertainties explicitly in the main content (e.g., Lines 481–499).

L263: This is a speculative statement that is not examined as thoroughly as it should be. If the sensitivity of Df to measurement uncertainty is taken into consideration is it still possible to conclude that the Df calculated here is 'relatively lower' than the previous measurements? If this result stands, can the authors provide any evidence to support their claim that fuel quality is higher in Beijing than for the studies they have cited?

Reply: The extBC mass determined at a given dmob is expected to have ~20% uncertainty as presented in the revision. The uncertainties should be similar at different mobility size. Thus, the Dfm, calculated as the scaling exponent of the mass-mobility relationship should influenced relatively low by the uncertainty in extBC mass. I mean the synchronous variations in extBC masses due to uncertainty at different dmob should have few effects on the Dfm. In the revision, we also examine the sensitivity of Dfm to the dmob range in which the power law function is used to fit the mass-mobility relationship. The Dfm fitted in 140–350 nm dmob range (2.51) is significantly larger than that fitted in 350–750 nm (2.07). It indicates the smaller extBC particles are more likely influenced by the reconstruction effect than the larger extBC (Lines 433–440).

The fuel quality is very strict in Beijing. The JING VI standard was implemented from year of 2017. We have no explicit information of the fuel quality. However, at least, we know the sulfate content is very low in the fuel with JING VI standard, which will have an important impact on the structure of BC aggregates. Besides, the local government claimed that the nitrogen oxides, particulate matters, total hydrocarbons and carbon monoxides emitted from the diesel exhausts are expected to decrease by factors of
**4.6%, 9.1%, 8.3% and 2.2% respectively if the fuel with JING VI standard is used.**

L337: 'extBC' as defined with respect to delay time could also be comprised of BC mixed with small or even moderate amounts of non-BC material (e.g. L280). Additionally, the measurements reported here are lower than previous DMA-SP2 measurements of effective density in Beijing. Therefore, I find this explanation of the low effective density values measured in this study to be problematic.

Reply: We have removed the discussion in diurnal variation in Dfm due to the enlarged uncertainty of insufficient data volume in the revision. Instead, we discuss the differences in the mass-mobility relationship between a polluted episode and a clean period. Although a fraction of the thinly- and/or even moderately-coated BC might be recognized as extBC using the time delay discrimination, these particles should have few effects on the determined typical extBC mass at each dmob. Explicit discussions have been presented in our revised manuscript (Lines 331-365). Increase in the volatile and/or semi-volatile materials accompanied with the BC aggregates would increase the possibility of the BC aggregates becoming relatively more compact due to reconstruction effect. It will increase the mass, consequentially the effective density of extBC aggregates at a given dmob. Even the structure of a BC aggregate changes little, small amounts of the volatile and/or semi-volatile materials filled in the gap and/or thinly-coated outside the primary spherules of BC aggregate would have few effects on its mobility but result in a higher mass of the entire particle. The increased mass can be detected by the APM but cannot be characterized by the SP2. Similar explanation is also presented in Lines 515–518 in the revision.

The effective density of internally mixed BC (In-BC, or thickly-coated BC) was measured by the previous DMA-SP2 system (Zhang et al., 2016). These In-BC particles have been reconstructed due to the thick coating. Thus, a compact structure with much higher density was reported. The fractal-like extBC aggregates without thick coating are studied in current study, which have low effective density.
References Gao, R. S., Schwarz, J. P., Kelly, K. K., Fahey, D. W., Watts, L. A., Thompson, T. L., Spackman, J. R., Slowik, J. G., Cross, E. S., Han, J.-H., Davidovits, P., Onasch, T. B., and Worsnop, D. R.: A novel method for estimating light-scattering properties of soot aerosols using a modified single-particle soot photometer, Aerosol Sci. Technol., 41, 125–135, doi:10.1080/02786820601118398, 2007.

Gysel, M., Laborde, M., Olfert, J. S., Subramanian, R., and Gröhn, A. J.: Effective density of Aquadag and fullerene soot black carbon reference materials used for SP2 calibration, Atmos. Meas. Tech., 4, 2851–2858, doi:10.5194/amt-4-2851-2011, 2011.

Laborde, M., Mertes, P., Zieger, P., Dommen, J., Balternsperger, U., and Gysel, M.: Sensitivity of the Single Particle Soot Photometer to different black carbon types, Atmos. Meas. Tech., 5, 1031–1043, doi:10.5194/amt-5-1031-2012, 2012.

Park, K., Cao, F., Kittelson, D. B., and McMurry, P. H.: Relationship between particle mass and mobility for diesel exhaust particles, Environ. Sci. Technol., 37, 577–583, doi:10.1021/es025960v, 2003.

Raatikainen, T., Brus, D., Hooda, R. K., Hyvärinen, A.-P., Asmi, E., Sharma, V. P., Arola, A., and Lihavainen, H.: Size-selected black carbon mass distributions and mixing state in polluted and clean environments of northern India, Atmos. Chem. Phys., 17, 371–383, doi:10.5194/acp-17-371-2017, 2017.

Sorensen, C. M.: The mobility of fractal agregates: A review, Aerosol Sci. Technol., 45, 765–779, doi:10.1080/02786826.2011.560909, 2011.

Taylor, J. W., Allan, J. D., Liu, D., Flynn, M., Weber, R., Zhang, X., Lefer, B. L., Grossberg, N., Flynn, J., and Coe, H.: Assessment of the sensitivity of core/shell parameters derived using the single particle soot photometer to density and refractive index, Atmos. Meas. Tech., 8, 1701–1718, doi:10.5194/amt-8-1701-2015, 2015.

Zhang, Y. X., Zhang, Q., Cheng, Y. F., Su, H., Kecorius, S., Wang, Z. B., Wu, Z. J., Hu, M., Zhu, T., Wiedensohler, A., and He, K. B.: Measuring the morphology and density
of internally mixed black carbon with SP2 and VTDMA: new insight into the absorption enhancement of black carbon in the atmosphere, Atmos. Meas. Tech., 9, 1833–1843, doi:10.5194/amt-9-1833-2016, 2016.

Please also note the supplement to this comment: https://www.atmos-meas-tech-discuss.net/amt-2018-408/amt-2018-408-AC2supplement.pdf

AMTD

**Supplement:**

1 *Supplement of*

**A study of the morphology and effective density of externally mixed black carbon aerosols in ambient air using a size-resolved single-particle soot photometer (SP2)**

Yunfei Wu et al.

*Correspondence to*: Yunfei Wu (wuyf@mail.iap.ac.cn) and Renjian Zhang (zrj@mail.iap.ac.cn)

8    Contents

29

[Figure]

30

**Fig. S1.** Temporal variation of the total number concentration of the scattering and incandescence particles detected by the SP2 during a size selection comprising of one short cycle with a duration of 18 s for each size and one long cycle with a duration of 36 s. The local peak at the beginning of each cycle was previously identified to correct the time difference between the DMA size selection and the SP2 measurement.

36

37

[Figure]

38

**Fig. S2.** Frequency distribution of the lag times between the incandescence and scattering peak

locations determined using the SP2. A bimodal distribution is found, with the minimum at ~2 μs.

The BC-containing particles with a lag time greater than 2 μs were considered to be thickly coated.

Otherwise, the BC-containing particles were non/thinly coated.

43

[Figure]

44

**Fig. S3.** The calibration curves for the relationship between the SP2-measured incandescence peak
height (low gain) and the refractory BC (rBC) mass. The Aquadag particles were used as the
representative material of rBC in the calibration. The Aquadag particles with different mobility
sizes ($d_{mob}$=100, 125, 150, 175, 200, 250, 300, 350, 400, 450 nm, respectively) were manually
selected by a DMA. The masses of the size-selected Aquadag particles were calculated based on
the effective densities ($\rho_{eff}$) according to the density-mobility relationship provided in Gysel et al.
(2011). The manual calibrations were performed both before (on 23 Jan. 2018) and after the
campaign (on 10 Feb. 2018). The red up-triangles represent the relationship between the mass of
Aquadag and the incandescence peak height during the calibration before campaign. The blue
down-triangles represent the calibration after the campaign. It is clear that the SP2 incandescence
peak height is proportional to the mass of Aquadag, inferring the linear regression functions as
y=0.003048x+0.1058 ($R^2$=0.9991) for the calibration before campaign and y=0.002972x+0.1713
($R^2$=0.9992) for the calibration after campaign. In the functions, y represents the mass of Aquadag,
and x represents the incandescence peak height. To examine the detection efficiency of SP2, the
size-resolved Aquadag particles were also automatically selected by the DMA in the $d_{mob}$ range of
140–750 nm and then measured using the SP2 and CPC after the campaign. The
mass-incandescence relationship derived from this measurement was very close to the manual
calibrations. The linear regression function is y=0.003012x+0.1097 ($R^2$=0.9999). The similar rBC
mass calibration curves indicate that the SP2 operated very stably throughout this campaign.
Considering that the incandescence signal peaks of Aquadag were ~25% higher than those of the
ambient BC particles, the slopes of the calibration equations should multiply by a factor of ~1.33
when converting the measured incandescence signal peaks to the rBC masses for ambient
particles.

[Figure]

68

69 **Fig. S4.** Detection efficiency of the SP2 as a function of (a) the mobility diameter of Aquadag and
70 (b) the mass-equivalent diameter of rBC. The detection efficiency was calculated as the ratio of
71 the number concentration of particles detected by the SP2 to that parallel measured by the CPC at
72 prescribed mobility sizes. Since the Aquadag particles were generated by an aerosol generator,
73 pure scattering particles should not exist. The particles having detectable incandescence signals
74 were recorded by the SP2. It is clear that adequate detection efficiencies (>~90%) for the SP2
75 were observed when the $d_{mob}$ of Aquadag was larger than 125 nm. Correspondingly, the
76 mass-equivalent diameter of rBC calculated using the $d_{mob}$ and $\rho_{eff}$ of Aquadag should be larger
77 than ~90 nm to provide adequate detection efficiencies. The relationship between the detection
78 efficiency of the SP2 and the rBC mass can be characterized using an exponential function, as
79 shown in Fig. S4b.

80

[Figure]

81

**Fig. S5.** Frequency distribution of the $d_{me}$ of $ext$BC at 16 prescribed mobility sizes (in the 140–750 nm $d_{mob}$ range) analyzed in the current study. Three thresholds of delay time between the incandescence signal peak and the scattering peak detected by the SP2 were employed to discriminate the $ext$BC particles, e.g., 2.0 µs, 1.2 µs and 0.4 µs. The frequency distributions corresponding to the thresholds are shown in red, green and blue colors, respectively. Reducing the delay time threshold results in a decrease in the data volume used in the statistics, while it seems to have few effects on the peak locations of the distributions. The major peak of the frequency distribution at smaller mobility sizes with a $d_{mob}<140$ nm is not clear due to the lower limit of the SP2. Thus, these smaller sizes were not discussed in the current study.

91

[Figure]

92

**Fig. S6.** The same as Fig. S5, but for the normalized number size distribution (d$N$/dlog$D_c$). The quantity $D_c$ indicates the mass-equivalent diameter of the rBC core, the same as $d_{me}$ defined above. Compared to Fig. S5, these graphs more clearly show that reducing the time-delay threshold has few effects on the peak $d_{me}$. The main changes occur at the right tail of the distributions at each mobility size. Reducing the time-delay threshold results in a significant reduction of particles at the right tail of the number size distribution. These particles are likely to be the thinly or even moderately coated BC particles, which were also recognized as *ext*BC using the time-delay approach. Since the peak $d_{me}$ values are needed in the current study, and are considered the typical $d_{me}$ values for the prescribed $d_{mob}$, these thinly or even moderately coated BC particles should have few effects on our results presented in this study. The peak $d_{me}$ value is identified as the mode value of a lognormal function that is fitted to the major peak of size distribution at each mobility.

105

[Figure]

106

107 **Fig. S7.** Time series of hourly PM$_{2.5}$ mass concentrations measured at the same site during the
108 experimental period. The dashed line represents the mean PM$_{2.5}$ mass concentration with a value
109 of 23.0±26.7 μg m$^{-3}$. The red and blue shaded regions represent a polluted episode (mean
110 PM$_{2.5}$=72.1±23.1 μg m$^{-3}$) and a consecutively clean period (mean PM$_{2.5}$=8.9±2.7 μg m$^{-3}$),
111 respectively, in which the mass-mobility relationships of *ext*BC are compared.
112

[Figure]

113

**Fig. S8.** The mass of *ext*BC particles as a function of the mobility diameter in the range of 140–750 nm (black circles) in the polluted (red up-triangles) and clean (blue down-triangles) episodes. The power-law functions are fitted to the mass-mobility relationships.

117

[Figure]

118

**Fig. S9.** The number fraction of *ext*BC as a function of the mobility diameter ($d_{mob}$) in the polluted
episode (red up-triangles) and clean period (blue down-triangles). The fraction of *ext*BC is roughly
calculated as the ratio of the *ext*BC number concentration to the sum of *ext*BC and internally
mixed BC (*int*BC) particles at each $d_{mob}$. The effect of multicharged particles is not eliminated.
However, the multicharged particles should result in a similar effect on *ext*BC and *int*BC at a
given mobility. Thus, there should be few effects on the number fraction of *ext*BC. A detailed
analysis of the mixing states of size-resolved BC particles will be performed in our future studies.

126

127  **Table S1.** The typical mass-equivalent diameters ($d_{me}$) and corresponding masses of *ext*BC at
128  different mobility sizes ($d_{mob}$) selected by the DMA in the whole campaign. Three delay-time
129  thresholds of 2.0, 1.2, and 0.4 µs are employed to discriminate *ext*BC.

| $d_{mob}$ (nm) | $d_{me}$ (nm) | | | mass (fg) | | |
|---|---|---|---|---|---|---|
| | $t_{lag} < 2.0$ µs | $t_{lag} < 1.2$ µs | $t_{lag} < 0.4$ µs | $t_{lag} < 2.0$ µs | $t_{lag} < 1.2$ µs | $t_{lag} < 0.4$ µs |
| 140 | 88.8 | 88.4 | 86.7 | 0.66 | 0.65 | 0.61 |
| 160 | 97.5 | 97.6 | 96.2 | 0.87 | 0.88 | 0.84 |
| 180 | 106.2 | 106.6 | 105.0 | 1.13 | 1.14 | 1.09 |
| 200 | 115.6 | 116.2 | 114.2 | 1.46 | 1.48 | 1.40 |
| 225 | 127.9 | 128.8 | 125.8 | 1.97 | 2.01 | 1.88 |
| 250 | 140.5 | 141.6 | 138.1 | 2.62 | 2.67 | 2.48 |
| 280 | 155.8 | 156.9 | 152.9 | 3.56 | 3.64 | 3.37 |
| 315 | 172.6 | 173.6 | 169.4 | 4.85 | 4.93 | 4.58 |
| 350 | 188.2 | 189.2 | 184.7 | 6.28 | 6.38 | 5.94 |
| 400 | 207.4 | 208.6 | 204.8 | 8.41 | 8.55 | 8.10 |
| 450 | 226.4 | 228.0 | 224.1 | 10.94 | 11.17 | 10.61 |
| 500 | 243.8 | 245.4 | 241.2 | 13.65 | 13.92 | 13.23 |
| 560 | 262.6 | 265.4 | 262.0 | 17.06 | 17.62 | 16.96 |
| 630 | 283.2 | 286.1 | 285.3 | 21.42 | 22.07 | 21.88 |
| 700 | 305.1 | 307.9 | 309.8 | 26.76 | 27.52 | 28.03 |
| 750 | 319.6 | 322.3 | 323.4 | 30.76 | 31.57 | 31.87 |

130

---

## Author Response (AR2)

The authors have responded comprehensively to the first round of reviews. The manuscript is greatly improved and I believe that it could be published in ACP once the following comments have been addressed.

We are deeply grateful to the reviewer for his/her positive comments to our revised manuscript. The two anonymous reviewers in the first round of review are also greatly appreciated for providing the constructive comments which have helped us to improve the quality of our paper.

The level of English language writing is still an issue. There are a large number of grammatical errors. I recognize that the authors have made efforts to improve this, including the hiring of professional language editing help. However, the level is simply still too low. There are still far too many grammatical errors to list in a scientific review. I recommend that another round of professional copy editing is required.

Response:

Thanks for the comments. Another round of professional copy editing has been performed to our revised manuscript before it is submitted back to the journal. The changes are marked with different colors in the revision.

The authors have scaled their Aqauadag calibrations by a size-independent factor to make them more similar to fullerene calibrations, which are thought to be more representative of ambient BC. The use of a size-independent scaling factor may introduce error since the ratio of Aquadag to fullerene peak heights in the SP2 is size dependent (Laborde et al., 2012). To avoid this problem, Baumgardner et al., (2012) introduced a single-point scaling procedure. I would recommend the authors also use this approach to make their results consistent with other SP2 studies.

Response:

Thanks for the professional comments. We have recalculated the calibration curve for the incandescence signal of our SP2 using the single-point scaling procedure provided by Baumgardner et al. (2012), i.e., scaling the peak high of broadband incandescence signals for Aquadag with 300 nm mobility diameter (i.e., 8.9 fg rBC mass) by a factor

of 0.75, and axis intercept at zero. The derived calibration factor is 0.0039, very close to the value (0.0040) we used in the current manuscript. This deviation (~2.5%) is even lower than that (~3%) during the calibrations performed pre- and post-campaign. The negligible difference in the estimated calibration factor should influence few on the results presented in the current manuscript. If the calibration factor is changed, there should be a massive data processing but has few impacts on the final results. Thus, we retain the original method in our current paper and add a sentence to express the similar calibration factor derived from the two method (Lines 284-287).

The authors have done a good job of including extra information in the revised supplementary information that increases confidence in their measurements.

The campaign was conducted over 19 days but only very limited time-resolved measurements are presented (one comparison between a clean period and a polluted period, which is very interesting). This seems like a missed opportunity. Were there any interesting variations in Dfm or rho\_eff over time? Even if not this would still be interesting to know. Is it possible to add a Figure that shows the time series of Dfm and rho\_eff at given mobility diameters? Perhaps also with the PM2.5 time series shown in Fig. S7 to see if there is any correlation. One of the advantages of the tandem DMA-SP2 technique is that it can perform measurements at relatively high time resolution, it seems a shame not to use this advantage.

**Response:**

Thanks for providing the useful comments. Actually, the initial purpose of this experimental setup was to investigate the mixing state of size-resolved BC-containing particles at a high time resolution, as well as the morphology and effective density of the uncoated BC aggregates. The results presented in this paper is one part of the whole study on the properties of size-resolved BC particles. We had tried to investigate the temporal variations in the morphology and effective density of the uncoated BC aggregates during the data processing. Unfortunately, there have no sufficient data volume to provide the stability and reliable size distribution from which the typical rBC

mass for a given mobility size was derived at a high time resolution, even on the daily scale. As shown in the attached figure, even for the two compared periods presented in our paper (i.e., a clean period and a polluted episode) with relatively adequate time, the data volume is still not sufficient enough as expected to obtain a wonderful number size distribution of the mass-equivalent diameter of the rBC core of *ext*BC. There are obvious fluctuations in the size distribution especially at larger mobility sizes (Fig. R1). Thus, we roughly discussed the differences in the mass-mobility relationship of extBC between the clean period and polluted episode in our current manuscript, although we know the time-resolved mass-mobility relationship should be more interesting. The low data volume is related to the low particle numbers at a certain mobility size and the low  $PM_{2.5}$  concentration in this campaign (23 µg m-3 on average).

Fig. R1 Number size distributions of the mass-equivalent diameter of the rBC core of *ext*BC normalized by the peak value at five represented mobility diameters (140, 225, 350, 500 and 750 nm) during a clean period (left) and a polluted episode (right). Lognormal fitting is performed for the major peak of each distribution.

The focus of this manuscript is on externally mixed BC particles (the revised manuscript now contains good, solid arguments for how the authors have isolated these particles). I think it would also be interesting to present the campaign-averaged Dfm and rho\_eff of internally mixed BC particles (i.e. those displaying delay time greater than 2 microseconds). This will require extra calculations and I know it is outside the main focus of the manuscript. But I think its important to put the extBC results in

context, which is the main focus of the manuscript. For example, is it actually true that the extBC particles display low Dfm because they are uncoated and more aggregatelike? This could be partially answered by checking if the coated particles that were present at the same time displayed higher Dfm.

**Response:**

The morphology and effective density of internally mixed BC were studied in the previous literature by using a VTDMA-SP2 system (Zhang et al., 2016). In our study, the DMA-SP2 is not likely to be used to study the morphology and effective density of internally mixed BC particles since the mobility diameter of these internally mixed BC particles were selected while only the mass of rBC core were determined by the SP2. Even the mass of the coating material can be estimated using the LEO fitting method and assumed density, larger uncertainties will be induced. Therefore, the DMA-SP2 system provide an advantage to study the mass-mobility relationship of externally mixed BC particles. For internally mixed BC particles, a VTDMA-SP2 system is required. As presented in Zhang et al. (2016), the average effective density of internally mixed BC particles measured at a suburban site nearby Beijing was 1.2 g/cm3, significantly higher than the values of externally mixed BC particles. Correspondingly, the internally mixed BC particles had a lower shape factor than the externally mixed ones. It means the internally mixed BC particles have a more compact structure. Zhang et al. (2016) also showed that the effective density of internally mixed BC particles was increased with the relative coating-thickness (Dp/Dc) perhaps due to the reconstruction of BC aggregates during their aging processing in the atmosphere.

Zhang, Y. X., Zhang, Q., Cheng, Y. F., Su, H., Kecorius, S., Wang, Z. B., Wu, Z. J., Hu, M., Zhu, T., Wiedensohler, A., and He, K. B.: Measuring the morphology and density of internally mixed black carbon with SP2 and VTDMA: new insight into the absorption enhancement of black carbon in the atmosphere, Atmos. Meas. Tech., 9, 1833–1843, doi:10.5194/amt-9-1833-2016, 2016.

Specific comments:

L 46: Higher rho\_eff values than what? This is an ambiguous statement. I guess it is meant higher than one might expect based on the trend observed at lower mobility diameters. But this is a poorly defined reference point.

**Response:**

We have changed the expression to make the meaning there clearer and more readable. (Lines 46–48)

L 104: What is meant by the term 'virtual Df'? Please clarify.

**Response:**

The 'virtual  $D_{\rm f}$ ' means the 'fractal dimension' defined as the scaling exponent between the radius of gyration ( $R_{\rm g}$ ) which is a root mean square radius that quantified the overall size of the aggregate, and the radius of primary spherules (*a*) composing the aggregates, expressed as:

$$\mathbf{N} = k_0 (\frac{R_g}{a})^{D_f},$$

where *N* is the number of primary paritcles and  $k_0$  is the scaling prefactor. We have added similar expression in the revision (Lines 106–108).

L 108: This sentence needs rewriting to clarify that it is the parameter effective density that is difficult to characterize by TEM, not the tandem measurements. (At least I think this is the intended meaning of the sentence.)

Response:

Modified (Lines 113–114).

L 113: Change to 'system detection limit' Response: Corrected (Line 117). L 121: 'Extrapolation' is a better and more specific word to use than 'applicability' here. Response:

Modified (Lines 125–127).

L 421: Are these extBC masses averaged over the full campaign? If so this needs to be explicitly stated.

**Response:**

Yes, the extBC masses presented here are the campaign average values. We have explicated it in the revision (Line 437).

L 535: Did any of the DMA-(APM, ELPI) studies use a thermodenuder to remove volatile components? This would not necessarily successfully remove all volatile material (as indicated on L 564 for the Rissler study). But it would be good to indicate if one was used or not here (as is done for the Rissler study), to indicate how much volatile material one might expect to be present in these previous measurements. Response:

In the previous literature for the study on the mass-mobility relationship of BC aggregates using the tandem method, the thermodenuder was seldom employed to remove volatile components because previous studies mostly focused on the BC aggregates freshly emitted from the diesel exhaust. The mass fraction of volatile components in these freshly emitted BC aggregates was considered generally low. The thermodenuder might have been used in some other DMA-(APM, ELPI) studies we missed to mention. However, the thermodenuder was not used in any of the studies cited in our current manuscript, except for Risslar et al. (2014).

L 567: My interpretation of this observation (that the discrepancy between DMA-APM and DMA-SP2 measurements is smaller at larger diameters) is that the particles at larger diameters are actually less coated (hence SP2 mass would agree more closely with APM mass). The ~300 nm diameter at which the discrepancy starts to decrease is consistent with the shift from a line with Dfm of 2.51 to one with Dfm of 2.07 shown in Fig. 3

(the kinks in the curves in Figs. 3 and 4 both occur ~300 nm). I'm not sure which interpretation is correct but perhaps this alternative one should at least be mentioned. Response:

We think carefully of this interpretation and found that it is with lower likelihood. Results shown in this study focus on the externally mixed BC particles, i.e., BC particles without coatings or with thin coatings. Thus, the coating effect should be negligible for the SP2 measured extBC particles regardless of the mobility size. We ever suspected that whether the smaller diesel exhaust particles (e.g., <300 nm) measured were more likely to be coated by other components resulting in a higher mass or effective density determined by the DMA-APM system. However, there is no sufficient evidence to support this hypothesis. Moreover, as shown in Fig. S9, the number fraction of ambient extBC in the entire SP2 detectable BC-containing particles showed a minimum at ~300 nm mobility diameter. It means that BC particles in this size range are more likely to be affected and coated by other components than the smaller or larger BC particles. Even the extBC was discussed in this study, the extBC particles in this mobility size range should also be more likely to be affected by other components resulting in a more compact structure than those with smaller or larger mobility sizes. The more compact BC aggregates also resulted in a relatively constant dynamic shape factor in the 200–350 nm mobility diameter range (Fig. 5).

[revised manuscript text omitted]

size distributions between the short-duration cycle and long-duration cycle. Although 320

321 the durations of each size in the short cycle and long cycle are different (18 s vs. 36 s),

322 the time difference between the size selection and the measurement should be uniform

for adjacent short and long cycles. Setting an initial time difference and calculating the mean number and/or mass concentration of each particle size, the number and/or mass size distributions are obtained. Then, the correlation coefficients between the size distributions during short and long cycles are calculated. Changing the time difference gradually, we can obtain a set of correlation coefficients as functions of the time differences. The time difference resulting in the maximum correlation coefficient is considered the difference between the size selection and the measurement.

Since the detection efficiency of the SP2 decreases dramatically in the small particle range (Fig. S4), the size distributions of the SP2-detected particles are inadequate for further calculation of the correlation coefficients. Therefore, the former method was employed in the current study to identify the time difference between the size selection and the SP2 measurement. The latter method will be used to examine the time difference between the size selection and the AE51/CPC measurements in our future study on the number and mass size distributions of BC.

337

**338 **3.2** Determination of the typical masses of extBC at prescribed mobility sizes**

Particles in a certain size range are selected by the DMA instead of absolutely 339 monodisperse particles at-in a given mobility size due to the effect of the transfer 340 341 function. In addition, larger particles with multiple charges are also selected. The 342 frequency and number size distributions of extBC as a function of the mass-equivalent 343 diameter of rBC ( $d_{me}$ ) at different mobility sizes are presented in Figs. S5 and S6, respectively. Note that the number size distribution has been normalized by the peak 344 value of the corresponding distribution. Since the frequency and number size 345 distributions of *ext*BC are quite insufficient at small particle sizes ( $d_{me} < 70$  nm) due to 346 347 the low detection efficiency of the SP2 (Fig. S4), only the distributions with a  $d_{\rm mob}$ 348 larger than 140 nm are presented. In our-the following study, we mainly address the morphology and effective density of extBC in the 140-750 nm dmob range. The 349 350 normalized number size distributions at five representative  $d_{\text{mob}}$  values (e.g., 140, 351 225, 350, 500, and 750 nm) are also shown in Fig. 2. The eExtBC particles having with 352 a considerable  $d_{\rm me}$  range were observed for a certain  $d_{\rm mob}$ , indicating a wide transfer

12

function of the DMA due to the relatively low ratio of sheath-to-sample flow (4.3:1). Multicharged particles also affected the size distribution, especially in the  $d_{\rm mob}$  range of 100–400 nm (Ning et al., 2013). As shown in Fig. S6 and Fig. 2, a minor peak is obviously observed at in the right tail of the major peak at each size distribution with afor  $d_{\rm mob}$  values of